# Mitochondria-associated membrane collapse is a common pathomechanism in *SIGMAR1*- and *SOD1*-linked ALS

Seiji Watanabe[1], Hristelina Ilieva[2,3], Hiromi Tamada[4], Hanae Nomura[1], Okiru Komine[1], Fumito Endo[1], Shijie Jin[1], Pedro Mancias[5], Hiroshi Kiyama[4] & Koji Yamanaka[1,*]

## Abstract

A homozygous mutation in the gene for sigma 1 receptor (Sig1R) is a cause of inherited juvenile amyotrophic lateral sclerosis (ALS16). Sig1R localizes to the mitochondria-associated membrane (MAM), which is an interface of mitochondria and endoplasmic reticulum. However, the role of the MAM in ALS is not fully elucidated. Here, we identified a homozygous p.L95fs mutation of Sig1R as a novel cause of ALS16. ALS-linked Sig1R variants were unstable and incapable of binding to inositol 1,4,5-triphosphate receptor type 3 (IP₃R3). The onset of mutant Cu/Zn superoxide dismutase (SOD1)-mediated ALS disease in mice was accelerated when Sig1R was deficient. Moreover, either deficiency of Sig1R or accumulation of mutant SOD1 induced MAM disruption, resulting in mislocalization of IP₃R3 from the MAM, calpain activation, and mitochondrial dysfunction. Our findings indicate that a loss of Sig1R function is causative for ALS16, and collapse of the MAM is a common pathomechanism in both Sig1R- and SOD1-linked ALS. Furthermore, our discovery of the selective enrichment of IP₃R3 in motor neurons suggests that integrity of the MAM is crucial for the selective vulnerability in ALS.

**Keywords** amyotrophic lateral sclerosis; inositol 1,4,5-triphosphate receptor type 3; mitochondria-associated membrane; sigma 1 receptor

**Subject Categories** Genetics, Gene Therapy & Genetic Disease; Neuroscience

## Introduction

Amyotrophic lateral sclerosis (ALS) is a progressive neurodegenerative disease that affects both upper and lower motor neurons, leading to death due to respiratory failure within a few years after the onset of the disease. Approximately, ~10% of ALS cases are inherited and dominant mutations in the gene for Cu/Zn superoxide dismutase (SOD1) are the second most frequent causes of inherited ALS (Cleveland & Rothstein, 2001; Bruijn *et al*, 2004). Mice overexpressing the *SOD1* gene with ALS-linked mutations recapitulate both the clinical and pathological characteristics of ALS. Although the precise mechanism of motor neuronal degeneration remains unclear, deregulated calcium ($Ca^{2+}$) homeostasis has been proposed as one of the key mechanisms (Grosskreutz *et al*, 2010; Tadic *et al*, 2014). For example, RNA editing on the α-amino-3-hydroxy-5-methyl-4-isoxazole propionate (AMPA) receptor subunit GluR2, which determines $Ca^{2+}$ permeability, is impaired in motor neurons of the cases with sporadic ALS (Takuma *et al*, 1999; Kawahara *et al*, 2004). In addition, a recent study has revealed that calpain, a $Ca^{2+}$-dependent protease, cleaves TAR DNA binding protein 43 (TDP-43), a protein abnormally accumulated in the lesion of sporadic ALS (Arai *et al*, 2006; Neumann *et al*, 2006), and induces the aggregation of TDP-43 (Yamashita *et al*, 2012). These studies suggest that disruption of $Ca^{2+}$ homeostasis is a critical factor involved in motor neuron degeneration in ALS.

Recessive mutations in *SIGMAR1* have recently been identified as a causative gene for ALS with or without frontotemporal lobar degeneration (FTLD) (Luty *et al*, 2010; Ullah *et al*, 2015), juvenile ALS (ALS16) (Al-Saif *et al*, 2011), and distal hereditary motor neuropathy (dHMN) (Li *et al*, 2015; Gregianin *et al*, 2016). Sigma 1 receptor (Sig1R), a gene product of *SIGMAR1*, is a chaperone protein highly expressed in spinal motor neurons (Mavlyutov *et al*, 2010) and specifically localized at an interface of the endoplasmic reticulum (ER) and mitochondria, called as mitochondria-associated membrane (MAM) (Hayashi & Su, 2007; Fujimoto & Hayashi, 2011). MAM regulates various functions including lipid metabolism (Vance, 2014), autophagy initiation (Hamasaki *et al*, 2013), and $Ca^{2+}$ transfer from ER to mitochondria via inositol 1,4,5-triphosphate receptors (IP₃R) and voltage-dependent anion channel 1 (VDAC1) (Hayashi & Su, 2007).

1 Department of Neuroscience and Pathobiology, Research Institute of Environmental Medicine, Nagoya University, Nagoya, Aichi, Japan
2 Houston Methodist Hospital, Houston, TX, USA
3 Department of Neurology, Johns Hopkins University, Baltimore, MD, USA
4 Department of Functional Anatomy and Neuroscience, Nagoya University Graduate School of Medicine, Nagoya, Aichi, Japan
5 Department of Pediatrics, The University of Texas Medical School at Houston, Houston, TX, USA
  *Corresponding author. Tel: +81 52 789 3865; Fax: +81 52 789 3891; E-mail: kojiyama@riem.nagoya-u.ac.jp

The recent studies suggest the role of MAM dysfunction in the ALS. Initially, altered subcellular localization of Sig1R was described in sporadic ALS cases (Prause *et al*, 2013). Subsequently, ER–mitochondria association, regulated by VAPB–PTPIP51 interaction, was found disrupted in various ALS cell models (De Vos *et al*, 2012; Stoica *et al*, 2014, 2016). Moreover, Bernard-Marissal and colleague have recently reported that a loss of Sig1R reduces ER–mitochondria cross talk and induces moderate motor neuron degeneration in Sig1R-deficient mice (Bernard-Marissal *et al*, 2015), indicating that Sig1R is important for integrity of the MAM. However, it is still controversial whether the loss of Sig1R function is responsible for the pathomechanism of *SIGMAR1*-linked ALS as evidence suggestive for gain of toxicity through Sig1R variants was recently provided. For instance, cytotoxicity was induced by overexpression of mutant Sig1R proteins in cultured cells (Tagashira *et al*, 2014; Gregianin *et al*, 2016), and elevated expression level of Sig1R in ALS and/or FTLD linked with *SIGMAR1* mutation in 3′-untranslated lesion was reported (Luty *et al*, 2010).

Here, we report that ALS16-linked Sig1R variants, including the novel p.L95fs mutation, are uniformly unstable and non-functional, indicating that a loss of function of Sig1R is causative for ALS16. Moreover, Sig1R deficiency induces dissociation of the MAM components and deregulation of $Ca^{2+}$ homeostasis at the MAM through mislocalization of $IP_3R$ type 3 ($IP_3R3$), resulting in calpain activation, mitochondrial dysfunction, and neurodegeneration. These results indicate that collapse of the MAM is critical both in Sig1R- and SOD1-linked ALS.

## Results

### Homozygous p.L95fs mutation in *SIGMAR1* is a novel cause of inherited juvenile ALS

Genetic testing was performed for a 20-year-old, juvenile onset ALS patient without an apparent family history (II-2, Fig 1A). She exhibited both upper and lower motor neuron involvement with slow disease progression. Details of the patient's history and examination were described in the Materials and Methods section. Since the initial genetic test was negative for spinal muscular atrophy, Charcot–Marie–Tooth disease, SOD1-linked ALS, spinocerebellar atrophy, and hereditary spastic paraplegia, whole-exome sequencing was performed and identified a homozygous c.283dupC, resulting in frameshift at position 95 (p.L95fs), mutation in the *SIGMAR1* gene (Fig 1B). Sequencing analyses of the *SIGMAR1* gene for both of biological parents revealed that the patient's father was heterozygous for this mutation, whereas her mother was negative. A genomewide single nucleotide polymorphism (SNP) genotyping for homozygosity mapping was performed on the DNA sample from the patient and her father. Homozygosity mapping revealed the complete absence of heterozygosity for chromosome 9 of the patient, where the *SIGMAR1* gene was located, indicating that the homozygous c.283dupC mutation in this patient is a result of paternal uniparental disomy for chromosome 9. These results support the idea that a homozygous frameshift mutation, p.L95fs, in *SIGMAR1* is a novel cause of inherited juvenile ALS.

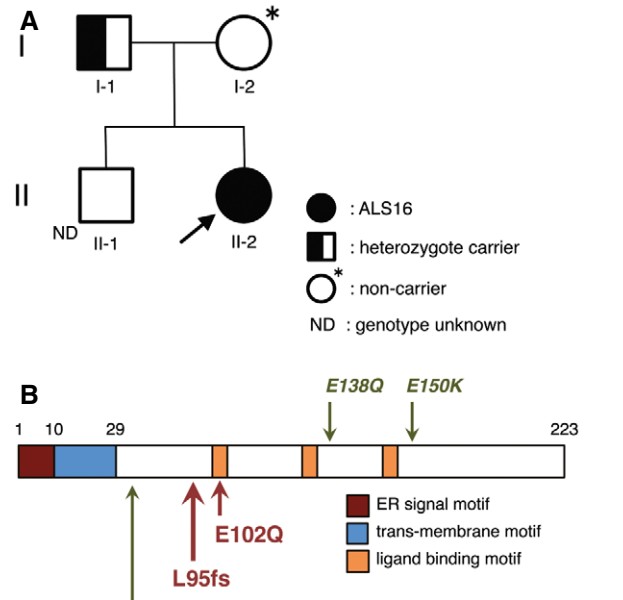

**Figure 1. p.L95fs is a novel amyotrophic lateral sclerosis (ALS) causative mutation in *SIGMAR1*.**

A   A family tree for the juvenile ALS patient with L95 fs mutation in sigma 1 receptor (Sig1R). The arrow indicates a proband.

B   Schematic diagram of domain structure of Sig1R, coded by *SIGMAR1*. Amino acid sequence misplaced by L95fs mutation is shown by a red rectangle. Note that both ALS-causative mutations (L95fs, E102Q; highlighted with red arrows) are located near the ligand-binding motif. In addition, recently identified mutations that cause distal hereditary motor neuropathy (dHMN), E138Q, E150K, and a splice-site mutation resulting in in-frame deletion of 20 amino acids (Δ31–50) are indicated by green arrows.

### ALS-linked Sig1R variants are unstable and nonfunctional

To characterize ALS-linked Sig1R variants, E102Q and L95fs, we overexpressed these Sig1R variants in N2a cells. Both of the variants were expressed at lower levels in the cells compared to the wild type (WT) (Fig 2A) and had shorter half-lives measured by the cyclohex-imide chase assay (Fig 2B). Treatment with proteasome inhibitor MG-132, but not with lysosomal protease inhibitors E64d/pepstatin A, significantly increased the amount of ALS-linked Sig1R variants in the cells (Fig 2C), suggesting their constitutive degradation via proteasomes. A recently identified Sig1R variant (p.32_50del) (Fig 1B), which is causative for autosomal recessive dHMN, is also unstable (Li *et al*, 2015), suggesting that Sig1R function is crucial for the maintenance of motor neurons. To determine the functional alteration of Sig1R at the MAM in ALS16, we fractionated the MAM as described before (Fig 2D) (Wieckowski *et al*, 2009; Fujimoto & Hayashi, 2011). We also established a cell line stably expressing human $IP_3R$ type 3 ($IP_3R3$), a binding partner of Sig1R (Hayashi & Su, 2007) (N2a-$IP_3R3$), because N2a cells did not express $IP_3R3$. $IP_3R3$ was enriched at the MAM both in N2a-$IP_3R3$ cells (Fig 2E) and in HeLa cells, which endogenously express $IP_3R3$ (Fig 2F). In contrast, endogenous $IP_3R$ type 1 ($IP_3R1$), a pan-neuronal $IP_3R$ subtype, did not localize at the MAM either in N2a-$IP_3R3$ cells

(Fig 2E) or in HeLa cells (Fig 2F). Co-immunoprecipitation assay revealed that E102Q Sig1R lost its ability to bind to IP$_3$R3 (Fig 2G). To assess the involvement of Sig1R deficits in intracellular Ca$^{2+}$ homeostasis, we suppressed the expression of Sig1R using siRNA against Sig1R (Fig 2H). N2a-IP$_3$R3 cells showed a more robust response of Ca$^{2+}$ release both in the cytoplasm and mitochondria on ATP stimuli than N2a cells, indicating that N2a-IP$_3$R3 cells are suitable for testing Ca$^{2+}$ homeostasis via Sig1R and IP$_3$R3 (Fig 2I and J). It should be noted that mitochondrial Ca$^{2+}$ flux was significantly affected by Sig1R knockdown only in N2a-IP$_3$R3 cells (Fig 2J). In the N2a-IP$_3$R3 cells, WT Sig1R successfully suppressed excess increase in Ca$^{2+}$ flux into the cytoplasm under knockdown of endogenous Sig1R, whereas both the E102Q and L95fs variants failed (Fig 2K). Conversely, only WT Sig1R increased mitochondrial Ca$^{2+}$ flux (Fig 2L). Thus, the ALS-linked Sig1R variants completely lost their function to control Ca$^{2+}$ flux. All these data indicate that the loss of function of Sig1R is a common mechanism of ALS16-linked ALS.

### Loss of Sig1R accelerates onset of the disease in SOD1$^{G85R}$ mice

Considering that ALS-linked SOD1 mutations provoke dysfunction both in mitochondria and ER, we hypothesized that Sig1R function at the MAM is also disrupted in SOD1-linked ALS. To elucidate the role of Sig1R in SOD1-linked ALS, we crossbred SOD1$^{G85R}$ and Sig1R$^{-/-}$ mice. In Sig1R heterozygous (Sig1R$^{+/-}$) mice, the level of Sig1R decreased to about half of the WT (Sig1R$^{+/+}$) (Fig 3A), but the onset and survival time of SOD1$^{G85R}$/Sig1R$^{+/-}$ mice were similar to those of SOD1$^{G85R}$ mice (Fig 3B). The onset of the disease of SOD1$^{G85R}$/Sig1R$^{-/-}$ mice remarkably accelerated by 65 days compared with SOD1$^{G85R}$ mice, while we observed modest retardation of the disease progression of SOD1$^{G85R}$/Sig1R$^{-/-}$ mice (duration of disease: 69 day, SOD1$^{G85R}$/Sig1R$^{+/+}$; 106 day, SOD1$^{G85R}$/Sig1R$^{-/-}$), resulting in the shortened survival time of SOD1$^{G85R}$/Sig1R$^{-/-}$ mice (Fig 3C). Consistent with the shortened total survival time, the body weight of mice decreased more rapidly in SOD1$^{G85R}$/Sig1R$^{-/-}$ mice than SOD1$^{G85R}$/Sig1R$^{+/-}$ or SOD1$^{G85R}$/Sig1R$^{+/+}$ mice (Fig 3D and E). The motor performance measured by rotarod tests also showed a faster decline in SOD1$^{G85R}$/Sig1R$^{-/-}$ mice older than 300 days (Fig 3F). On the other hand, Sig1R deficiency itself did not affect the viability by about 400 days of age (Fig 3B and C) and Sig1R$^{-/-}$ mice showed mild decline in motor performance measured by the rotarod tests with aging (Fig 3G and H), suggesting that the Sig1R deficiency alone causes only a modest motor phenotype in mice.

### Mutant SOD1 proteins are accumulated at the MAM and mitochondria in the cultured neurons and affected tissues

To investigate the mechanisms through which Sig1R deficiency exacerbated the disease of mutant SOD1 mice, we performed the subcellular fractionation from mutant SOD1-expressing N2a cells and SOD1 transgenic mouse tissues as shown in Fig 2D. In contrast to the cytosolic localization of SOD1$^{WT}$ protein, we observed mutant SOD1 was partially localized in the mitochondrial fraction in N2a cells and the spinal cords of end-stage mutant SOD1 mice as previously reported (Fig 4A and B; Liu *et al*, 2004). In addition, we found that mutant SOD1 was also accumulated at the MAM both in N2a cells and in the spinal cord of end-stage mutant SOD1 mice (Fig 4A and B). Intriguingly, the accumulation of mutant SOD1 at the MAM was also observed in the brain, but not in the liver or primary cultured astrocytes of SOD1$^{G93A}$ mice (Fig 4C). These findings indicate that accumulation of mutant SOD1 at the MAM is specifically observed in neurons and tissues affected in ALS, and is correlated with neurodegeneration. Moreover, mutant SOD1 prevented ER membranes from binding to mitochondria (Fig 4D), indicating the direct inhibition of the MAM association by mutant SOD1. The inhibition of ER–mitochondria association *in vitro* was dependent on mutant expression in mitochondria (Fig 4D, lane 7 and 8). This was consistent with the previous study that VDAC1 is one of the primary

**Figure 2. Both E102Q and L95fs Sig1R variants are unstable and lose their abilities to control Ca$^{2+}$ flux in Neuro2a cells.**

A    Immunoblotting analysis of Neuro2a (N2a) cells expressing Sig1R-FLAG variants.
B    Cycloheximide chase analysis on N2a cells expressing Sig1R-FLAG. Quantitative data of immunoblotting for the levels of Sig1R-FLAG protein and its variants during the cycloheximide chase were plotted as mean ± SEM of three independent experiments (upper panel). Representative immunoblots for the levels of Sig1R-FLAG proteins were shown (lower panel). **$P < 0.01$, *$P = 0.0216$ in E102Q versus wild type (WT); ##$P < 0.01$, #$P < 0.05$ in L95fs versus WT. A one-way ANOVA with subsequent *post hoc* Tukey's test.
C    N2a cells transfected with Sig1R-FLAG variants were incubated with MG-132 (10 μM) or the combination of E64d and pepstatin A (5 μg/ml each) (E64d/PepA) for 8 h. The relative mean levels of Sig1R-FLAG variants determined by immunoblotting from three independent experiments were plotted as mean ± SEM (upper panel). Representative immunoblots were shown (lower panel). *$P < 0.05$ versus no inhibitor control. A one-way ANOVA with subsequent *post hoc* Tukey's test.
D    Schematic representation of subcellular fractionation used in this study. Details are described in the Materials and Methods section (Cyto, cytosol; P1, nucleus and debris; Mito, mitochondria; MAM, mitochondria-associated membrane; P3, microsomal fraction).
E, F   Subcellular localization of inositol 1,4,5-triphosphate receptor type 1 (IP$_3$R1) and type 3 (IP$_3$R3) in N2a cells stably expressing human IP$_3$R3 (N2a-IP$_3$R3 cells, E) or HeLa cells (F). IP$_3$R1 and IP$_3$R3 in the isolated fractions were identified by immunoblotting with the specific antibodies as indicated. Proper fractionation of these samples was confirmed by the fraction-specific markers as indicated.
G    Interaction of IP$_3$R3 with wild-type Sig1R or E102Q ALS-linked Sig1R mutant. Sig1R-FLAG variants were transfected in N2a-IP$_3$R3 cells, and IP$_3$R3 was co-immunoprecipitated using an anti-FLAG antibody and identified by immunoblotting with the specific antibodies as indicated.
H    Suppression of Sig1R by siRNA. Lysates of N2a cells transfected with control siRNA (siCtrl) or siRNA against Sig1R (siSig1R) for 24 h were blotted. Similar results were obtained from three independent experiments. Paired *t*-test was used for statistical analysis.
I, J    Cytoplasmic (I) or mitochondrial (J) calcium (Ca$^{2+}$) flux in N2a or N2a cells stably expressing human IP$_3$R3 (N2a-IP$_3$R3). siCtrl or siSig1R was transfected into N2a or N2a-IP$_3$R3 cells, then cytoplasmic and mitochondrial Ca$^{2+}$ flux were determined with fluo-4 and Case12-mito, respectively. The fluorescence intensity was normalized to the intensity in resting state at 0 s and plotted as mean ± SD. **$P < 0.01$, *$P < 0.05$. Two-way ANOVA with subsequent *post hoc* Tukey's test. $n = 10$ each.
K, L    Cytoplasmic (K) or mitochondrial (L) Ca$^{2+}$ flux in N2a-IP$_3$R3 cells. siCtrl or siSig1R was transfected with or without the Sig1R-FLAG variants. The data are obtained and plotted as described in (I and J). $n = 10$ each. **$P < 0.01$, *$P < 0.05$. Two-way ANOVA with subsequent *post hoc* Tukey's test.

Source data are available online for this figure.

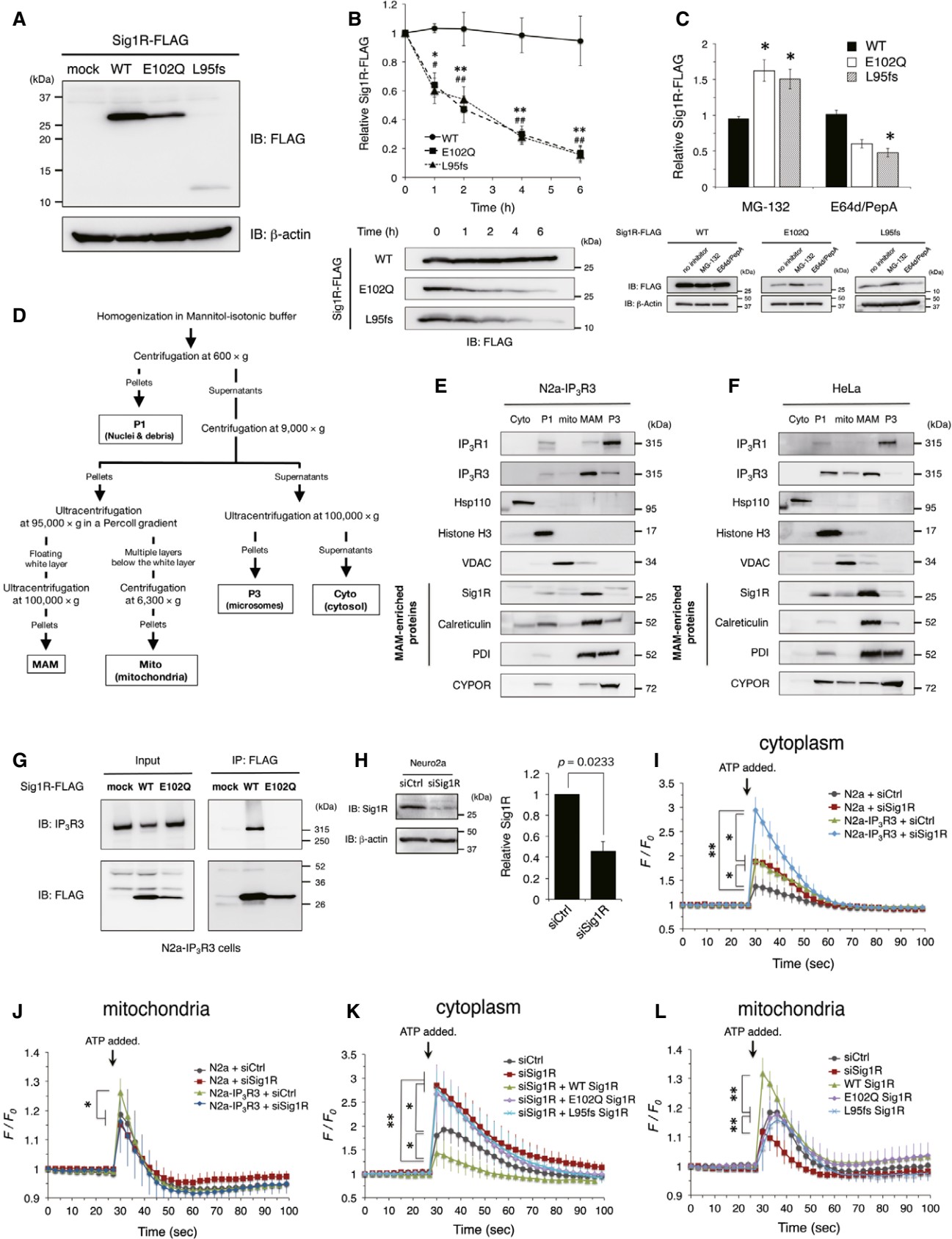

**Figure 2.**

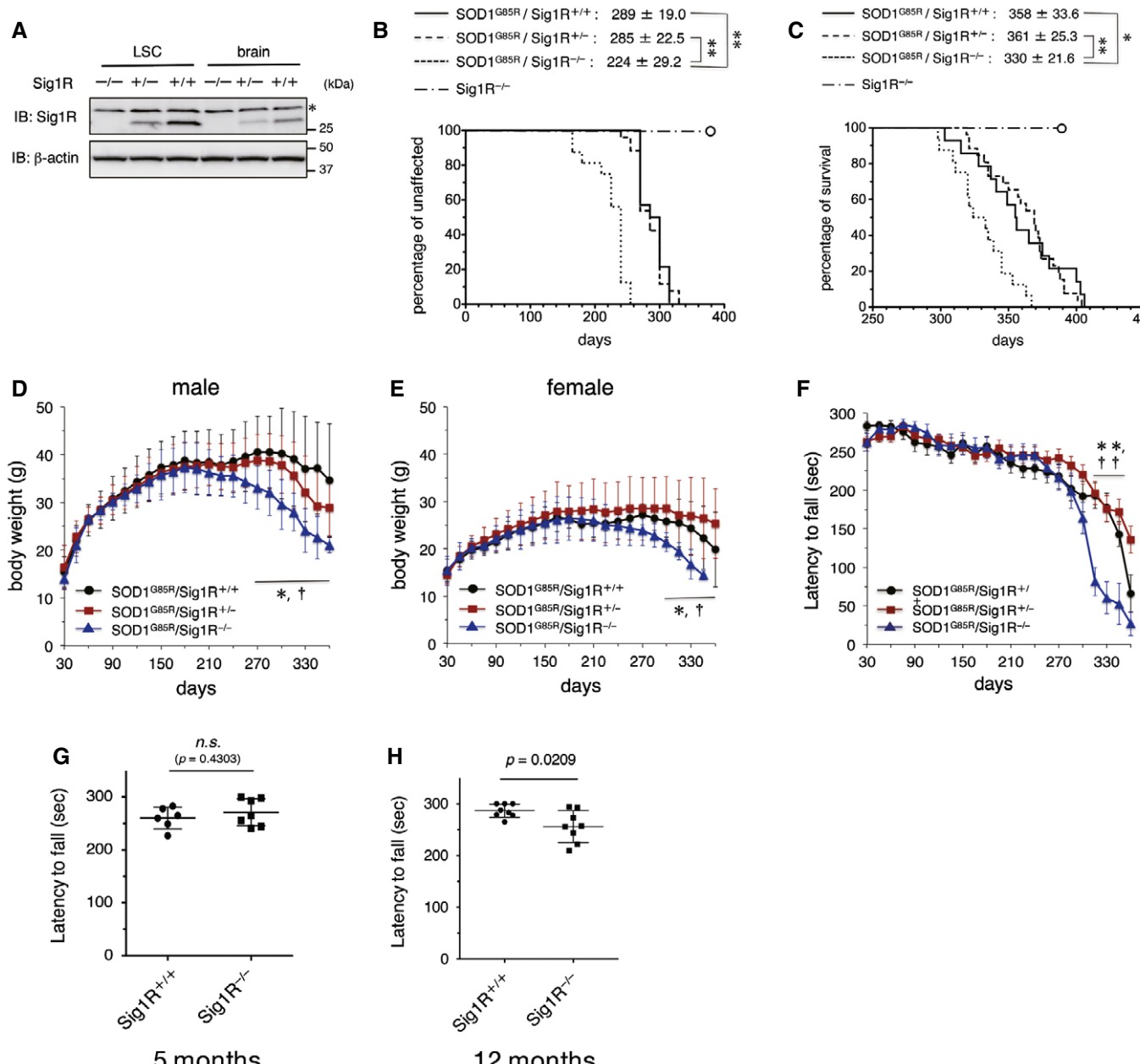

**Figure 3. Accelerated disease onset in Sig1R-deficient SOD1^G85R mice.**

A    The levels of Sig1R in the lumbar spinal cord (LSC) or brain of Sig1R⁻/⁻, Sig1R⁺/⁻, or Sig1R⁺/⁺ mice at 5 months old. Representative immunoblots obtained from three independent experiments are shown. Asterisk denotes non-specific bands.

B, C    Sig1R-deficient SOD1^G85R mice (SOD1^G85R/Sig1R⁻/⁻) exhibited accelerated onset of the disease (B) and shortened survival time (C) compared to SOD1^G85R mice with one or two copies of Sig1R (SOD1^G85R/Sig1R⁺/⁻, SOD1^G85R/Sig1R⁺/⁺). Open circles indicate Sig1R knockout mice (Sig1R⁻/⁻) that never developed motor neuron disease but that were sacrificed at 396 days of age (n = 8). Mean onset or survival times of mice with ± SD are shown on the top of Kaplan–Meyer curves. **P < 0.01, *P < 0.05; log-rank test. n = 14 (SOD1^G85R/Sig1R⁺/⁺), 29 (SOD1^G85R/Sig1R⁺/⁻), 17 (SOD1^G85R/Sig1R⁻/⁻).

D, E    Body weight was lost earlier both in male (D) and female (E) SOD1^G85R/Sig1R⁻/⁻ mice. Data are shown as mean ± SD. *P < 0.05 in SOD1^G85R/Sig1R⁺/⁺ versus SOD1^G85R/Sig1R⁻/⁻, †P < 0.05 in SOD1^G85R/Sig1R⁺/⁻ versus SOD1^G85R/Sig1R⁻/⁻. Two-way ANOVA with subsequent *post hoc* Tukey's test. n = 8 (SOD1^G85R/Sig1R⁺/⁺), 13 (SOD1^G85R/Sig1R⁺/⁻), 8 (SOD1^G85R/Sig1R⁻/⁻) in male (D); and n = 6 (SOD1^G85R/Sig1R⁺/⁺), 16 (SOD1^G85R/Sig1R⁺/⁻), 9 (SOD1^G85R/Sig1R⁻/⁻) in female (E). The animals were the same as those used in panels (B and C). Error bars denote SD.

F    Decreased performance on the rotarod test was observed in SOD1^G85R/Sig1R⁻/⁻ mice. The test was performed at 0–30 rpm with 0.1 rpm/s acceleration every week. Data are shown as mean ± SD. **P < 0.0001 in SOD1^G85R/Sig1R⁺/⁺ versus SOD1^G85R/Sig1R⁻/⁻, ††P < 0.0001 in SOD1^G85R/Sig1R⁺/⁻ versus SOD1^G85R/Sig1R⁻/⁻. Two-way ANOVA with subsequent *post hoc* Tukey's test. The animals were the same as those used in panels (B–E). Error bars denote SD.

G, H    Motor function of WT (Sig1R⁺/⁺) and Sig1R knockout (Sig1R⁻/⁻) mice at 5 or 12 months old was evaluated by the rotarod test (0–30 rpm, accelerated at 0.1 rpm/s). Average time on the rotating rod was plotted. Error bars denote SD. Unpaired *t*-test.

Source data are available online for this figure.

targets of mutant SOD1 on mitochondrial outer membrane (Israelson *et al*, 2010). It should also be noted that both Sig1R and IP$_3$R3 were not directly bound to mutant SOD1 (not shown). Accumulation of SOD1$^{G93A}$ at the MAM was observed at a pre-symptomatic stage but temporarily reduced around the onset of the disease (Fig 5A and B). A similar time course was observed in SOD1$^{G85R}$ (Fig 5C), while reduction in SOD1$^{G37R}$ level at the MAM was observed after the disease onset (Fig 5D). These findings suggest that disruption of the MAM in the spinal cord neurons might be responsible for the disease onset and/or perhaps progression. On the other hand, other MAM-specific proteins, such as Sig1R, IP$_3$R3, and calreticulin, continuously disappeared from the MAM fraction without degradation as the disease progressed (Fig 5E). Of note, Sig1R was accumulated in mitochondrial fractions rather than the MAM when we fractionated neural tissues of the mutant SOD1 transgenic mice (Fig EV1C–E). These observations suggest that the MAM structure was dissociated in mutant SOD1 transgenic mice.

## IP$_3$R3 is mislocalized from the MAM in the presence of mutant SOD1 or the absence of Sig1R

To elucidate the role of the Sig1R–IP$_3$R3 interaction on motor neuron degeneration, we first examined the expression of Sig1R and IP$_3$R3 in the adult central nervous system (CNS). Sig1R was widely expressed in the CNS, whereas IP$_3$R3 expression was restricted to the spinal cord and brainstem regions containing motor neurons such as the anterior horn of the spinal cord (Fig 6A), the hypoglossal nucleus (Fig 6B), and the motor nuclei of the facial and trigeminal nerves (Fig EV2). IP$_3$R3 expression was not detected in hippocampus (Fig 6C), cerebral cortex, or cerebellum (Fig EV2). Sig1R and IP$_3$R3 were co-localized in all the neurons expressing both the molecules (Fig 6A and B). To analyze changes in Sig1R and IP$_3$R3, SOD1$^{G93A}$ mouse spinal cords were immunostained in a pre-symptomatic stage (Fig 6D) and end stage (Fig 6E). In both stages, Sig1R formed aggregates predominantly in anterior horn neurons as previously reported (Prause *et al*, 2013), and IP$_3$R3 lost its normal localization at the MAM and was diffused throughout the cell body. Moreover, co-localization of Sig1R and IP$_3$R3 disappeared in SOD1$^{G93A}$ lumbar motor neurons. Importantly, Sig1R aggregation and IP$_3$R3 diffusion started from the pre-symptomatic stage (Fig 6D). Similar observations were obtained in end-stage SOD1$^{G85R}$ (Fig 6F) and SOD1$^{G37R}$ (Fig 6G) mice, but not in SOD1$^{WT}$ mice (Fig 6H). These data imply that the disruption of the Sig1R–IP$_3$R3 interaction is involved in selective degeneration of motor neurons. To clarify whether the mislocalization of IP$_3$R3 was dependent on Sig1R, we also examined this in Sig1R$^{-/-}$ mouse spinal cord and found that IP$_3$R3 similarly disappeared from the MAM (Fig 6I). Since Sig1R deficiency did not change subcellular localization of protein disulfide isomerase, a MAM-enriched ER marker, in motor neurons (Fig EV3A), a loss of Sig1R does not cause robust alteration in ER morphology. To examine structural alteration of the MAM in detail, we also evaluated the MAM in motor neurons of SOD1$^{G85R}$, Sig1R$^{-/-}$, or non-transgenic control mice by electron microscopy (Figs 6J and EV3B). Approximately 11% of the mitochondrial surface associated with ER in the control motor neurons (Fig 6K), which was well consistent with the previous study (Stoica *et al*, 2014). On the other hand, both in the SOD1$^{G85R}$ and Sig1R$^{-/-}$ motor neurons, the ER–mitochondria

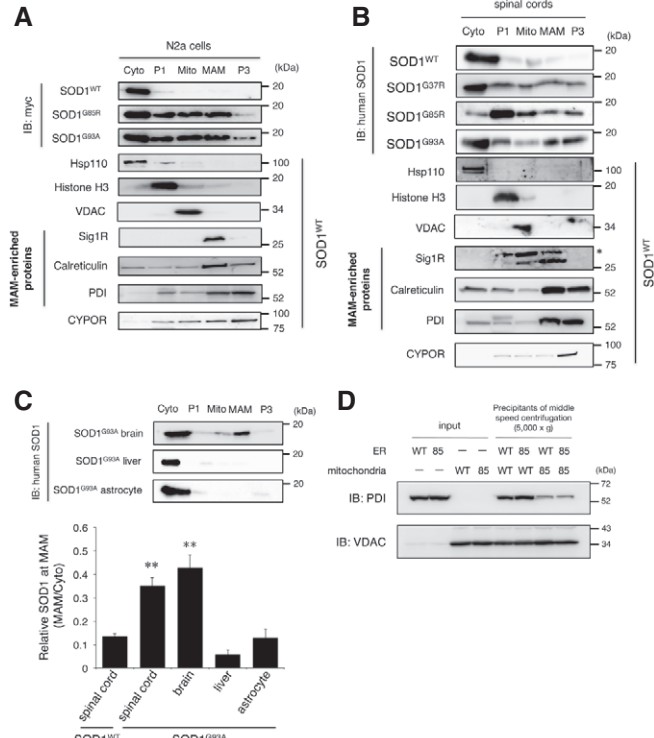

**Figure 4.  Mutant SOD1 proteins accumulate at the MAM and mitochondria in mice and cells expressing mutant SOD1.**

A, B  N2a cells expressing SOD1$^{WT}$, SOD1$^{G85R}$ or SOD1$^{G93A}$ (A), or spinal cords of WT or mutant SOD1 transgenic mice at end stage (B) were fractionated as described in the Materials and Methods section (Cyto, cytoplasm; P1, nuclei and debris; Mito, mitochondria; MAM, mitochondria-associated membrane; P3, microsomal fraction). Proper fractionation was confirmed by the fraction-specific markers as indicated and also shown in Fig EV1A–E. The asterisk denotes non-specific bands.

C  Tissue- and cell type-specific accumulation of mutant SOD1 at the MAM. Brain, liver, or primary astrocytes from SOD1$^{G93A}$ mice were fractionated and blotted as in (A and B) (upper panel). Quantification of SOD1 proteins at the MAM relative to ones in the cytoplasm was performed from immunoblotting data with an anti-human SOD1 antibody in (B) and the top panel of (C), and the data are plotted as mean ± SEM from three independent experiments (lower panel). **$P$ < 0.01 versus SOD1$^{WT}$ spinal cord; one-way ANOVA with subsequent *post hoc* Tukey's test. Proper fractionation of these samples was confirmed in Fig EV1F–H.

D  Mutant SOD1 prevented association of ER with mitochondria. Isolated ER (P3: microsomal fraction) and mitochondria of N2a cells expressing wild-type (WT) or G85R (85) SOD1 for 48 h were mixed and incubated. Mitochondrial pellets after centrifugation were analyzed by immunoblotting using anti-PDI (ER marker) and anti-VDAC (mitochondrial marker), respectively. All these results were confirmed by three independent experiments.

Source data are available online for this figure.

contacted area was apparently decreased (~8.2 and 7.5%, respectively) (Fig 6J and K), indicating that the ultrastructure of the MAM was impaired in these mice. Furthermore, the amounts of IP$_3$R3 and calreticulin, another MAM-enriched protein, substantially decreased from the MAM fractions without degradation in Sig1R$^{-/-}$ mice (Fig 6L), supporting the notion that Sig1R is crucial for integrity of the MAM.

## Disruption of the Sig1R–IP$_3$R3 interaction leads to deregulated Ca$^{2+}$ homeostasis, calpain activation, and the reduction in ATP synthesis

The previous studies showed that Ca$^{2+}$ flux from ER to mitochondria was important for ATP synthesis in mitochondria (Fujimoto & Hayashi, 2011) and that calpain activated by Ca$^{2+}$ release into the cytoplasm was involved in ALS-linked pathological TDP-43 processing (Yamashita *et al*, 2012). In line with the cited studies, the

expression of IP$_3$R3 in N2a cells exacerbated mutant SOD1-mediated toxicity (Fig 7A), suggesting that Ca$^{2+}$ deregulation at the MAM induced by mislocalization of IP$_3$R3 is involved in the motor neuron degeneration. To reveal the mechanism through which disruption of the MAM leads to neurodegeneration, we measured the levels of intracellular ATP and calpain activation under such conditions. In N2a-IP$_3$R3 cells, overexpression of Sig1R did not affect basal calpain activity (Fig 7B); however, WT Sig1R successfully suppressed calpain activation induced by mutant SOD1 (Fig 7C and D).

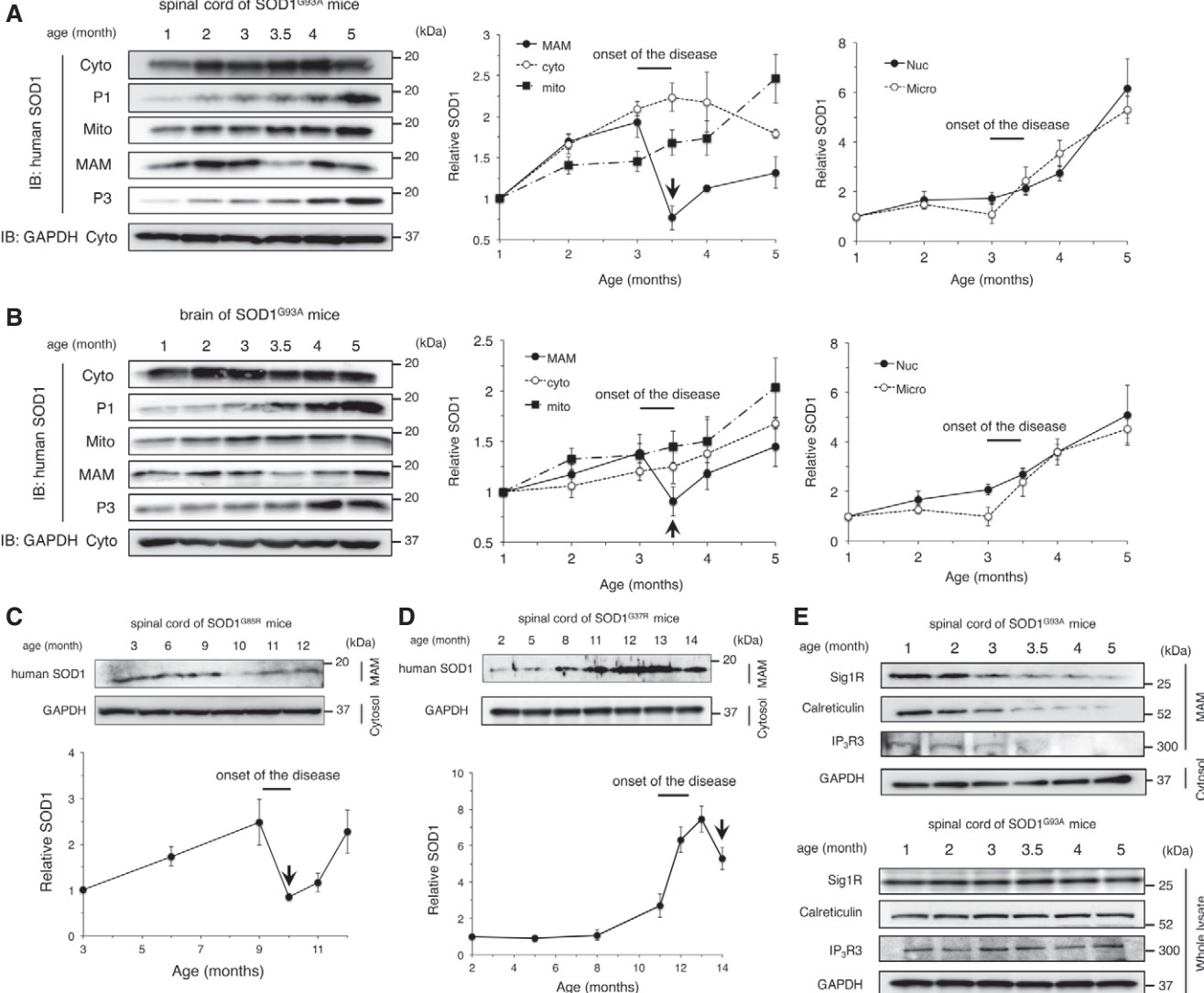

**Figure 5. Accumulation of mutant SOD1 at the MAM is most prominent around the disease onset.**

A, B   Time course analyses of mutant SOD1 levels at the MAM in SOD1$^{G93A}$ mouse spinal cords (A) and brains (B). Immunoblots show levels of mutant SOD1 protein in the indicated fractions at various time points (left panels). Quantitative data at the right were plotted as mean ± SEM of three independent experiments. Arrows indicate a rapid reduction in SOD1 level at the MAM just following the onset of disease.

C, D   Time course analyses of mutant SOD1 accumulation at the MAM in SOD1$^{G85R}$ (C) and SOD1$^{G37R}$ (D) mouse spinal cords. Quantitative data from three independent experiments were plotted as mean ± SEM (bottom). Arrows indicate a rapid reduction in SOD1 level at the MAM just following the onset of disease.

E   Time course analyses of MAM-specific proteins at MAM and in the whole lysates of SOD1$^{G93A}$ mouse spinal cords.

Data information: Fractions from a single mouse tissue at the indicated ages were loaded on each lane. Representative blots from at least three independent experiments were shown. GAPDH in the cytosolic fractions was used as a loading control.

Source data are available online for this figure.

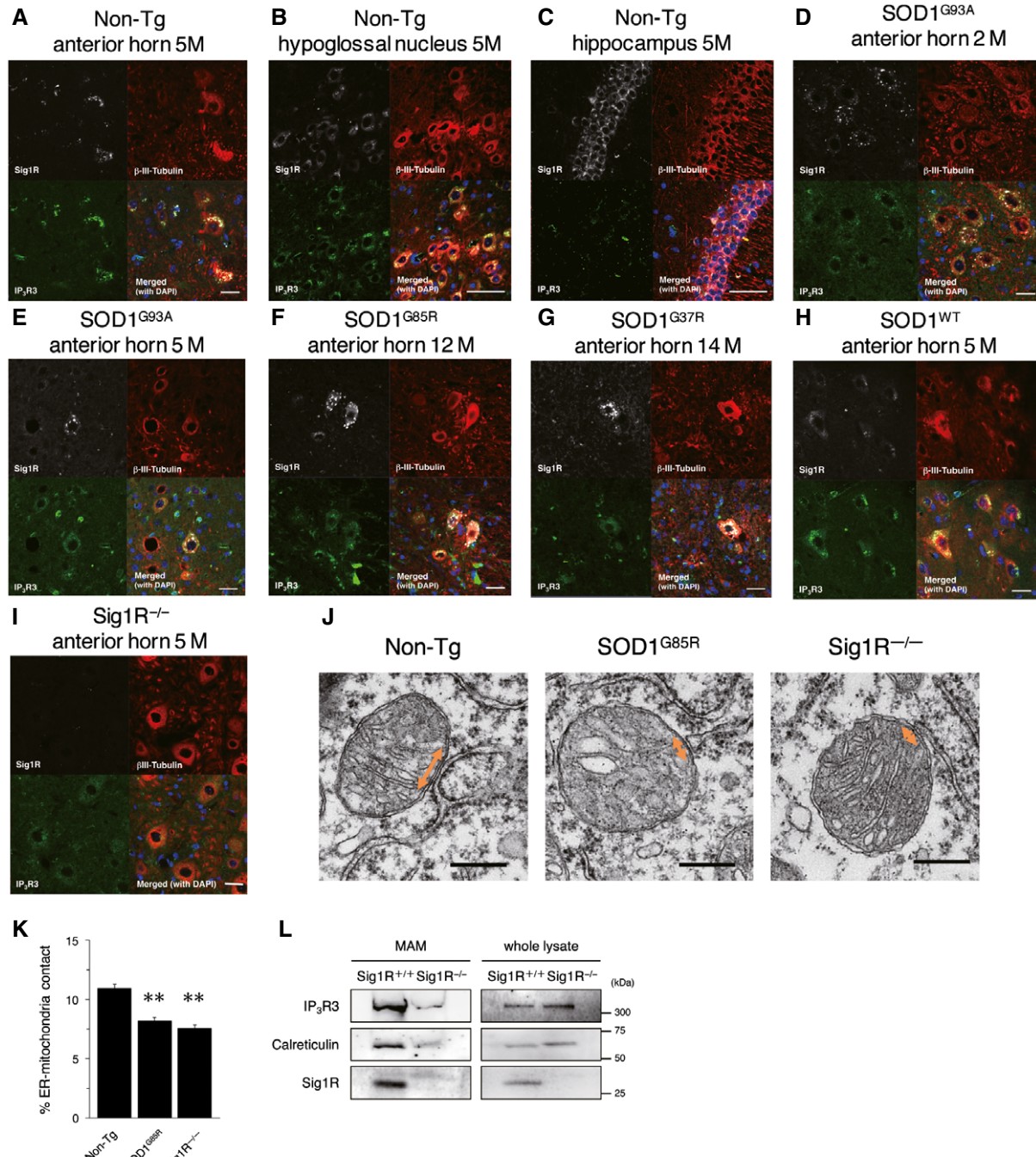

**Figure 6. Loss of Sig1R or expression of mutant SOD1 compromises the MAM integrity in motor neurons.**

A–I    Immunofluorescence staining of spinal cords and brains from non-transgenic (Non-Tg), SOD1 transgenic, or Sig1R$^{-/-}$ mice. Transverse sections of mouse spinal cords (A, D–I) or sagittal sections of mouse brains (B, C) were stained using anti-Sig1R (white), βIII-tubulin (red), and IP$_3$R3 (green) antibodies. Note that Sig1R and IP$_3$R3 are co-localized in the motor neurons of the anterior horn (A) and the hypoglossal nucleus (B), and IP$_3$R3 was not expressed in hippocampal neurons (C). Mutant SOD1 induced aggregation of Sig1R and mislocalization of IP$_3$R3 in anterior horn neurons (D–G), while their abnormalities were not observed in SOD1$^{WT}$ motor neurons (H). Mislocalization of IP$_3$R3 was also observed in Sig1R$^{-/-}$ mouse spinal cords (I). Scale bars: 50 μm.

J, K    Representative electron micrographs of the MAM (J) (double-headed arrows) in motor neurons of 12-month-old Non-Tg, SOD1$^{G85R}$, or Sig1R$^{-/-}$ mice. Note that ER–mitochondria contacted areas were reduced in both SOD1$^{G85R}$ or Sig1R$^{-/-}$ mice. Quantification of the mitochondria surface associated with ER was calculated in (K). For quantification, 13–19 motor neurons and 224–316 mitochondria with MAM in each animal (n = 2) were analyzed. Data are expressed as mean ± SEM. **P < 0.0001 versus Non-Tg; one-way ANOVA with subsequent *post hoc* Tukey's test. Scale bars: 300 nm.

L    The levels for IP$_3$R3 and calreticulin were decreased in the MAM fractions of Sig1R$^{-/-}$ mouse brains. MAM fractions and whole tissue lysates of Sig1R$^{+/+}$ or Sig1R$^{-/-}$ mouse brains were immunoblotted. Representative blots from three independent experiments are shown.

Source data are available online for this figure.

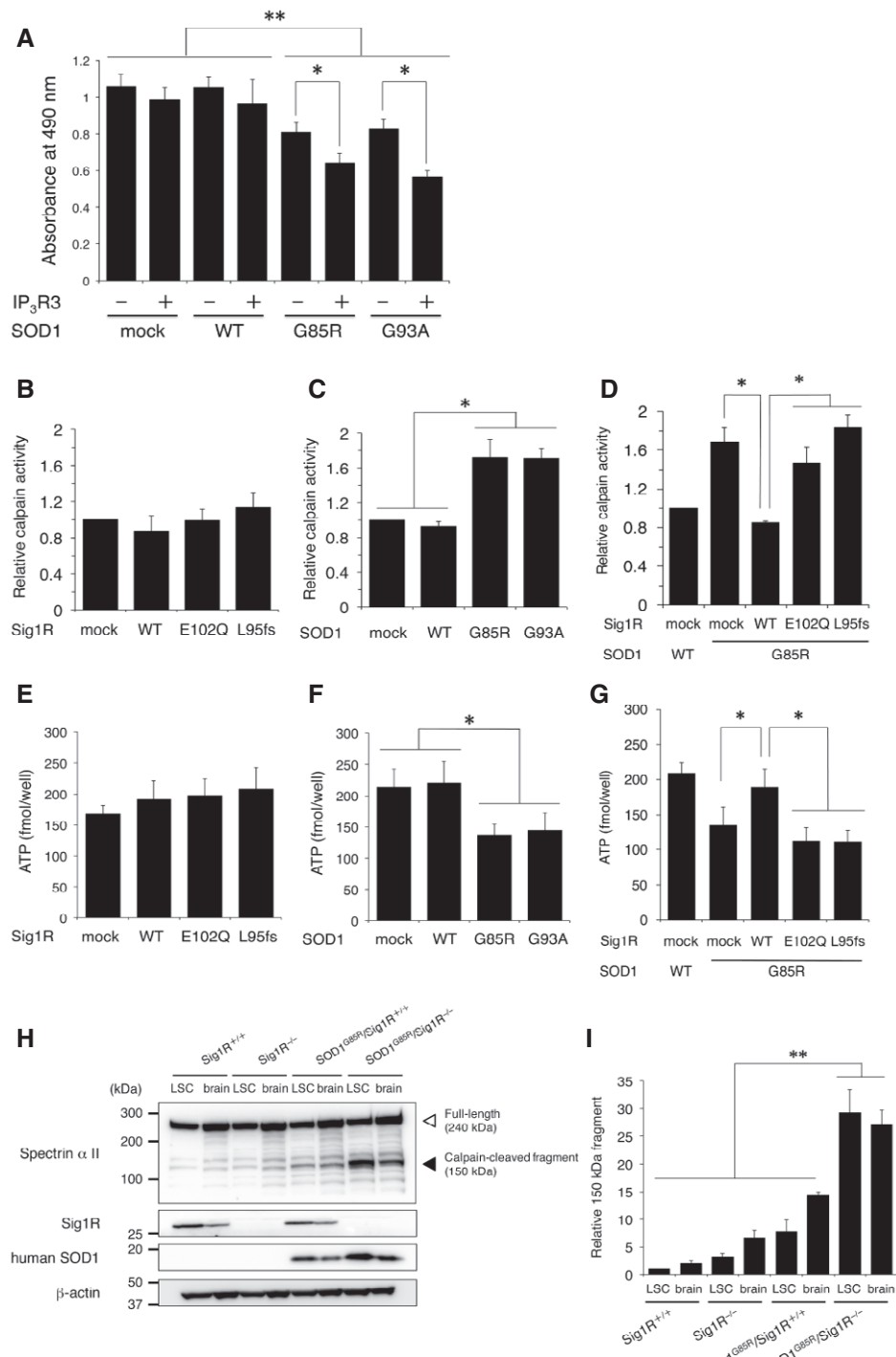

**Figure 7. Sig1R inhibits mutant SOD1-mediated, abnormal calpain activation and restores intracellular ATP levels.**

A   N2a cells were transfected with WT or mutant SOD1 in the presence or absence of IP$_3$R3. Plotted viability of the cells measured by a neurotoxicity assay revealed that IP$_3$R3 is involved in cell vulnerability against mutant SOD1. Data are expressed as mean ± SEM from three independent experiments, triplicated in each experiment.

B–G  Calpain activity (B–D) and intracellular ATP levels (E–G) were measured in N2a-IP$_3$R3 cells expressing Sig1R-FLAG (B and E), SOD1 (C and F), or both (D and G). Mean ± SEM from three independent experiments is plotted.

H, I  Calpain activity *in vivo* was determined by the cleavage of spectrin α II. Calpain-cleaved 150 kDa fragment of spectrin α II in mouse lumbar spinal cord or brain (H) was normalized to β-actin (I). Quantitative data in immunoblotting analyses are plotted as mean ± SEM from three independent experiments.

Data information: **$P < 0.01$, *$P < 0.05$; one-way ANOVA with subsequent *post hoc* Tukey's test (A–G, I).
Source data are available online for this figure.

Similarly, reduction in intracellular ATP level caused by mutant SOD1 was also recovered only by WT Sig1R (Fig 7E–G). In both cases, ALS16-linked Sig1R variants, E102Q and L95fs, lost their abilities to control calpain activation and ATP levels. To examine whether Sig1R deficiency affects calpain activation *in vivo*, we evaluated cleavage of spectrin α II, a major substrate of calpain (Higuchi *et al*, 2005). As shown in Fig 7H and I, calpain-mediated cleavage of spectrin α II fragments remarkably increased in the brain and lumbar spinal cord of SOD1$^{G85R}$/Sig1R$^{-/-}$ mice compared with SOD1$^{G85R}$, while, consistent with the modest decline of motor performance, there was a tendency toward an increase in calpain activation in Sig1R$^{-/-}$ mice (Fig 7H and I).

### PRE-084, a Sig1R agonist, restores the Sig1R–IP$_3$R3 interaction

As shown in previous studies (Hyrskyluoto *et al*, 2013; Ono *et al*, 2013), PRE-084, an agonist for Sig1R, showed a neuroprotective effect in mutant SOD1 transgenic mice. However, it was unclear whether PRE-084 affected the Sig1R–IP$_3$R3 interaction in motor neurons. To address this question, we evaluated Ca$^{2+}$ flux into the cytoplasm (Fig 8A) and mitochondria (Fig 8B), calpain activity (Fig 8C), and intracellular ATP levels (Fig 8D) in N2a-IP$_3$R3 cells in the presence of PRE-084 or NE-100, an antagonist for Sig1R. In all these experiments, PRE-084 successfully restored the function of IP$_3$R3 disrupted by mutant SOD1, suggesting that Sig1R activation by PRE-084 prevented the disruption of the Sig1R–IP$_3$R3 interaction. On the other hand, NE-100 failed to restore the function of Sig1R deregulated by mutant SOD1. Moreover, intraperitoneal administration of PRE-084 in pre-symptomatic SOD1$^{G93A}$ mice successfully restored co-localization of Sig1R and IP$_3$R3 in anterior horn neurons of the lumbar spinal cord (Fig 8E and F). All these data indicate that the Sig1R activation that regulates IP$_3$R3-mediated intracellular Ca$^{2+}$ flux at the MAM is crucial to prevent MAM disruption in motor neurons both *in vitro* and *in vivo*.

## Discussion

In this study, we demonstrated for the first time that two ALS-linked Sig1R variants, including the novel p.L95fs mutation, are unstable

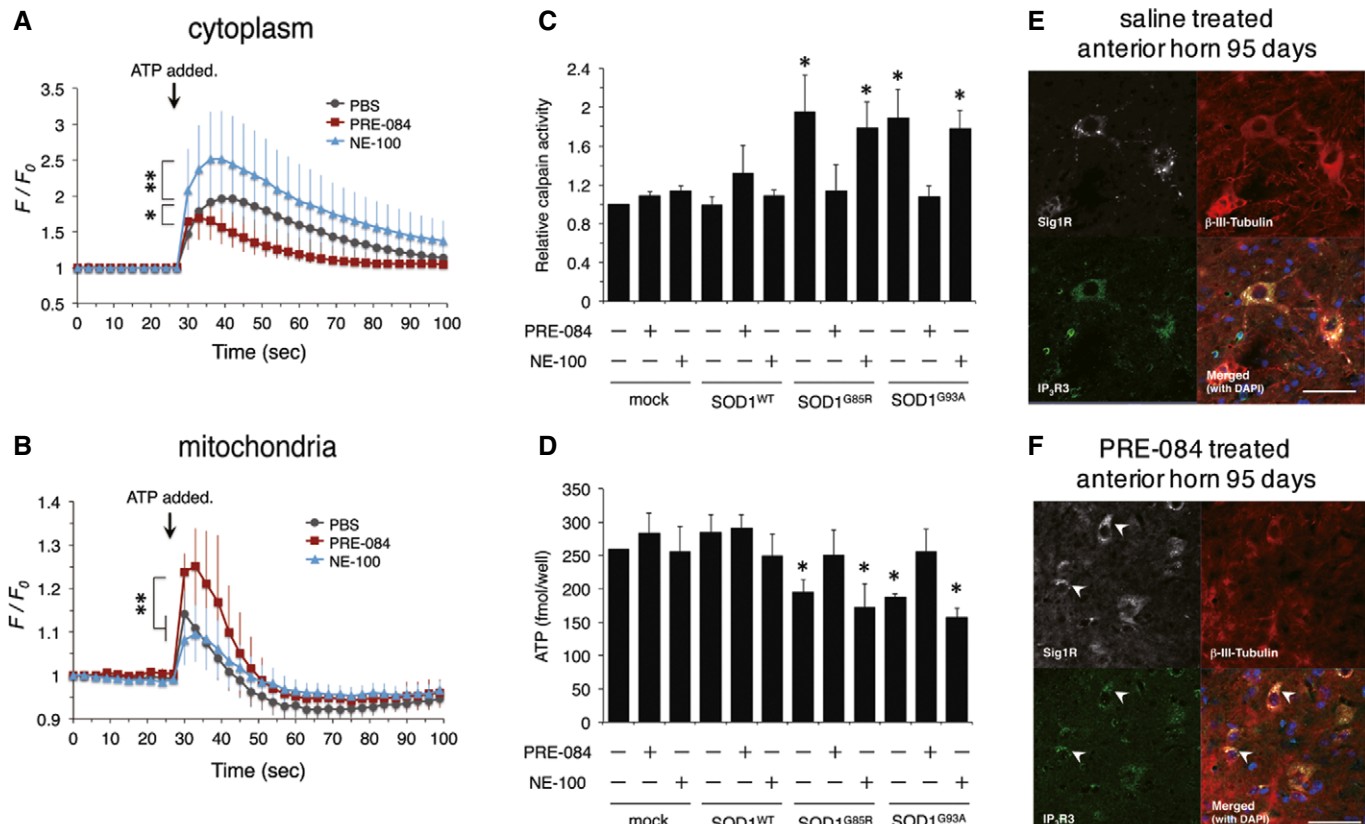

**Figure 8.  Pre-symptomatic administration of PRE-084, a Sig1R agonist, restores Sig1R functions at the MAM and prevents Sig1R aggregation *in vivo*.**

A, B  Cytoplasmic (A) or mitochondrial (B) Ca$^{2+}$ flux was measured in N2a-IP$_3$R3 cells treated with PRE-084 (5 μM) or NE-100 (5 μM) for 1 h prior to fluorescent imaging. Cytoplasmic and mitochondrial Ca$^{2+}$ flux were detected by fluo-4 and Case12-mito, respectively. The fluorescence intensity was normalized by the resting state at 0 s and plotted as mean ± SD. *n* = 10 each. **$P < 0.01$, *$P < 0.05$; two-way ANOVA with subsequent *post hoc* Tukey's test.

C, D  Calpain activity and intracellular ATP levels were measured in N2a-IP$_3$R3 cells treated with PRE-084 (5 μM) or NE-100 (5 μM) for 24 h. Data are plotted as mean ± SEM from three independent experiments. *$P < 0.05$ versus mock control; one-way ANOVA with subsequent *post hoc* Tukey's test.

E, F  SOD1$^{G93A}$ mice were intraperitoneally administered with saline or PRE-084 (0.25 mg/kg, 3 times per week) from postnatal day 35 to 95. Lumbar spinal cord sections were stained by using anti-Sig1R (white), IP$_3$R3 (green), and βIII-tubulin (red) at 95 days old. Arrowheads indicate the neurons with normal distribution of Sig1R and IP$_3$R3. Scale bars: 50 μm.

and lose their abilities to bind IP$_3$R3 and to control Ca$^{2+}$ homeostasis at the MAM. A recent crystallographic study of Sig1R revealed that a pair of hydrogen bonds formed by Glu102, Val36, and Phe37 in Sig1R tethers its cytosolic domain to the transmembrane domain (Schmidt et al, 2016), providing a structural explanation for instability of the ALS-linked Sig1R variants. Moreover, ALS-linked Sig1R variants were unable to control the Ca$^{2+}$ flux in Sig1R-depleted neuronal cells. Taken together, we concluded that the loss-of-function mechanism is mainly responsible for Sig1R-linked ALS.

Our results also indicate that a loss of Sig1R is sufficient for inducing collapse of the MAM in vivo for the first time. Electron microscopic study revealed dissociation of ER and mitochondria at the MAM in the motor neurons of Sig1R$^{-/-}$ mice. Although there are a number of the molecules implicated in controlling the interaction between ER and mitochondria at the MAM, the molecule responsible for the integrity of the MAM has not been identified. Sig1R is one of such key factors to maintain the structure and function likely through collaborating with other MAM proteins. In Sig1R$^{-/-}$ mice, we showed that Sig1R deficiency induced the dissociation of MAM components. In mutant SOD1 mice, the MAM was also disrupted through Sig1R aggregation. Therefore, the collapse of MAM is a common pathomechanism in both *SIGMAR1*- and *SOD1*-linked ALS. Considering the Sig1R aggregation was also observed in sporadic ALS cases (Prause et al, 2013) and the MAM disruption was similarly observed in VAPB (De Vos et al, 2012)-, TDP-43 (Stoica et al, 2014)-, or FUS (Stoica et al, 2016)-linked ALS cell models, dysfunction of the MAM might widely be involved in various ALS cases. Further studies are required to provide a complete picture of the role of MAM dysfunction in ALS.

Although it had been pointed out that mutant SOD1 proteins are bound to the mitochondrial outer membrane (Liu et al, 2004), their accumulation at the MAM was not clarified. Our study revealed for the first time that mutant SOD1 proteins also accumulated prominently at the MAM fraction in the spinal cords of mutant SOD1 mice. However, the significance of mutant SOD1 proteins accumulated at the MAM remains unclear. The MAM plays important roles not only in trafficking Ca$^{2+}$ into mitochondria but also in proteostasis. MITOL, an E3 ubiquitin ligase localized at the MAM, ubiquitinates and targets mutant SOD1 for degradation (Yonashiro et al, 2009), and autophagosome formation initiates at the MAM (Hamasaki et al, 2013). Since ER stress sensors, such as PERK and BiP, are enriched at the MAM (Fujimoto & Hayashi, 2011), mutant SOD1 accumulated at the MAM may trigger an ER stress response. These studies imply that mutant SOD1 proteins were accumulated at the MAM to trigger the intracellular proteostatic responses. Although the precise mechanism through which accumulation of misfolded SOD1 disrupts the MAM and forms aggregation of Sig1R remains unclear, one possible mechanism is that aberrant binding of mutant SOD1 to the outer membrane of mitochondria prevents the association of mitochondrial proteins with the ER. Indeed, the prevention of ER–mitochondria association by mutant SOD1 was dependent on mutant expression in mitochondria, and mutant SOD1 did not directly bind to both Sig1R and IP$_3$R3. Rapid reduction in the accumulated SOD1 levels in the MAM at or after the disease onset may also represent severe MAM disruption in motor neurons of mutant SOD1 mice. The SOD1$^{G85R}$/Sig1R$^{-/-}$ mice showed the acceleration of disease onset about 20% of the total survival time in a similar manner to SOD1$^{G93A}$/Sig1R$^{-/-}$ mice (Mavlyutov et al, 2013). This suggests that Sig1R deficiency predominantly affects the MAM integrity in motor neurons because motor neurons are primary determinant of disease onset (Boillee et al, 2006). On the other hand, the disease duration was slightly extended in the same mice. The change on disease progression suggests involvement of glial cells (Yamanaka et al, 2008a,b). Since the MAM also regulates an innate immune response (Fujimoto & Hayashi, 2011), one possible hypothesis is that MAM dysfunction in glial cells might partially reduce neurotoxic inflammation. A precise role of the MAM in glial cells remains to be an open question.

Mammals have three different types of IP$_3$R: IP$_3$R1, type 2 (IP$_3$R2), and IP$_3$R3. IP$_3$R1 is ubiquitously expressed in the CNS, whereas IP$_3$R2 is mainly expressed in glial cells, and IP$_3$R3 is expressed in the subset of developing neurons (Sharp et al, 1999; Blackshaw et al, 2000; Futatsugi et al, 2008). The co-localization of Sig1R and IP$_3$R3 in adult motor neurons, which we first revealed in this study, and the predominant localization of IP$_3$R3 at the MAM suggest that IP$_3$R3 is likely to be the key subtype responsible for selective vulnerability of motor neurons in ALS. Moreover, we revealed that mislocalization of IP$_3$R3 from the MAM induced calpain activation and mitochondrial dysfunction. Previous studies have proposed involvement of calpain in neurodegeneration (Higuchi et al, 2005; Yamashita et al, 2012). Calpastatin, an endogenous inhibitor of calpain, has been shown to be protective against SOD1$^{G93A}$-mediated toxicity in vitro (Stifanese et al, 2010). Intriguingly, activated calpain-1 damages lysosomal membranes and induces lysosomal impairment (Villalpando Rodriguez & Torriglia, 2013). Related to this, we recently reported that lysosomal impairment is also involved in neuronal death in the mutant SOD1 model (Watanabe et al, 2014b). Thus, IP$_3$R3-mediated excess cytoplasmic release of Ca$^{2+}$ is probably an initiation step in motor neuron degeneration via calpain. On the other hand, we also showed that a reduction in Ca$^{2+}$ flux into the mitochondria led to lower ATP content in neuronal cells. A deficit of Ca$^{2+}$ uptake results in reduced mitochondrial motility and neuronal loss in Huntington's disease mice (Panov et al, 2002), and Sig1R$^{-/-}$ or NE-100-treated neurons showed elongated mitochondria with lower motility in axons to cause axonal degeneration and progressive death of primary neurons (Bernard-Marissal et al, 2015).

It should be noted that Sig1R$^{-/-}$ mice exhibited mild motor phenotypes, whereas a loss of Sig1R function develops juvenile ALS (ALS16) in human. One possible explanation is that it may be difficult to reproduce the motor deficit within the short life span of mice due to a very slow disease progression of ALS16. Indeed, despite of their early onset (2–5 years old), only two out of six ALS16 patients were bound to wheelchairs by 20 years old (Al-Saif et al, 2011) and our ALS16 patient was still able to walk at the age of 20, whereas most sporadic ALS patients lost their lives within 1–5 years. On the other hand, Bernard-Marissal and colleagues reported relatively small but significant nerve denervation and loss of motor neurons in Sig1R$^{-/-}$ mice (Bernard-Marissal et al, 2015). In our study, age-dependent mild motor phenotype was observed by rotarod tests, and calpain activity tended to be increased in Sig1R$^{-/-}$ mice. These observations suggest that other triggering molecules such as mutant SOD1 are required to induce severe phenotypes in mice due to their much shorter life span than human. Despite of these facts, it should also be noted that our findings in Sig1R$^{-/-}$ mice such as MAM

disruption and Sig1R aggregation are well consistent with the pathological features reported in sporadic ALS cases in human (Prause et al, 2013), and therefore, Sig1R$^{-/-}$ mice are the useful tool to examine the molecular pathomechanism of ALS.

Bernard-Marissal and colleagues have recently reported that MAM dysfunction by Sig1R deficiency mediates motor neuron degeneration in mice. They showed that Sig1R deficiency concomitantly induced both ER stress and mitochondria alternation through Ca$^{2+}$ deregulation at the MAM using Sig1R$^{-/-}$ primary neurons. Their findings are well consistent with our results. In addition to this, we showed that calpain activation and reduced ATP synthesis were also induced by the MAM dysfunction in SOD1-ALS model and those phenotypes were aggravated when Sig1R is deficient. All these observations indicate that collapse of the MAM compromises various intracellular homeostatic pathways, and therefore, the MAM might be a promising target for the future therapy. In contrast to the cited work, our study is the first to point out that Sig1R deficiency clearly induced IP$_3$R3 mislocalization, and the loss of Sig1R–IP$_3$R3 interaction is responsible for the Ca$^{2+}$ deregulation at the MAM both in Sig1R- and SOD1-linked ALS.

Recent studies have reported the involvement of MAM dysfunction in various neurodegenerative diseases, such as Alzheimer's disease (AD) and Parkinson's disease (PD). In AD, presenilin-1 and presenilin-2, which processes the amyloid precursor protein (APP) to generate amyloid-β, predominantly localized at the MAM (Area-Gomez et al, 2009), and up-regulated MAM proteins were found in the AD brain and APP-transgenic AD mouse model (Area-Gomez et al, 2012; Hedskog et al, 2013). Therefore, in contrast to ALS studies including ours, over-activation of MAM function seems to be involved in neurodegeneration in AD, while parkin, DJ-1, and α-synuclein, the gene products of inherited PD, are required for neuroprotective function of the MAM and interaction with ER and mitochondria (Paillusson et al, 2016). In addition, mutant α-synuclein accumulated at the MAM and induces mitochondrial damage (Guardia-Laguarta et al, 2014). These observations suggest that deregulation of the MAM may be, at least in part, a pathomechanism common to the several neurodegenerative diseases including AD, PD, and ALS. Further studies to uncover the players and mechanisms of MAM deregulation are required to understand precise roles of the MAM in neurodegenerative diseases.

As we demonstrated, collapse of the MAM is a common pathomechanism in Sig1R- and SOD1-linked ALS. Depletion of MAM proteins and disruption of intracellular Ca$^{2+}$ homeostasis were found both in Sig1R$^{-/-}$ mice and in mutant SOD1 transgenic mice. Sig1R deficiency markedly accelerated the onset of the disease in SOD1$^{G85R}$ mice. Ca$^{2+}$ deregulation and calpain activation, which we demonstrated here, were also reported in TDP-43-linked disease models (Yamashita et al, 2012; Aggad et al, 2014). Moreover, mislocalization of aggregated Sig1R (Prause et al, 2013) and intracellular Ca$^{2+}$ deregulation (Grosskreutz et al, 2010; Tadic et al, 2014) has been also pointed out in sporadic ALS. All these observations imply that deregulation of Ca$^{2+}$ homeostasis at the MAM is a general mechanism of neurodegeneration in ALS. Our findings suggest that the deregulation of Ca$^{2+}$ homeostasis via IP$_3$R3 is one of the key mechanisms for motor neuron degeneration. These results provide us with new perspectives regarding future therapeutics, especially focused on preventing the MAM disruption for SOD1- and Sig1R-linked ALS patients, and perhaps, sporadic ALS patients.

# Materials and Methods

## Subjects

The affected patient from a non-consanguineous, Hispanic family was diagnosed with juvenile ALS at the University of Texas Medical School at Houston, Houston Texas, USA. The genetic studies for the patient and her parents were performed as a part of a diagnostic examination with informed consent obtained from each subject. Genomic DNAs were extracted from peripheral blood lymphocytes. After the genetic tests, the family had genetic counseling. The patient's brother and her parents did not show any neurological abnormalities. The use of genetic and clinical information in this study was approved by the institutional review board of Research Institute of Environmental Medicine, Nagoya University (approval number: # 347).

## Description of the patient

The patient was born via vaginal delivery without complications and met all developmental milestones until age 5 years when she was seen for toe walking. Within 1 year, she developed high steppage gait with mild bilateral foot drop and was found to have increased lower extremity muscle tone. Over the years, she experienced progressive muscle weakness with significant atrophy of distal muscles with development of pes cavus and wasting of calf muscles and intrinsic muscles of the hands. She experienced no bowel, bladder or swallowing difficulties. At the time of the last visit, she was 20 years old and 6-months pregnant and followed by a high-risk obstetrician. Neurological examination was notable for intact cranial nerves without dysarthria and tongue fasciculations. Muscle mass was normal except distally, where she has significant atrophy of the intrinsic hand muscles and distal leg wasting. She had a mild position and action hand tremor. Muscle strength was minimally affected proximally (5-/5), but significantly in the hands and feet (0/5 dorsiflexion, 3/5 plantarflexion). Deep tendon reflexes were 3+ in the upper extremities and knees and 4+ at the ankles with extensor plantar responses. Sensory examination was normal to all modalities. Coordination was intact except for poor fine motor skills. She had a high steppage, wide-based gait. MRI of the brain and spinal cord and SSEPs of the upper and lower extremities were all normal. CSF analysis was normal. Plasma levels of lactic acid, pyruvic acid, VLCFA, vit. E, folate, and vit. B12 were normal. Nerve conduction study showed normal sensory responses; compound muscle action potential showed moderately to severely reduced amplitudes, mildly prolonged duration of distal latency, and low normal conduction velocities in upper and lower limbs. EMG showed increased insertional activity with 3+ fibrillations distally, none proximally. Motor units showed chronic denervation distally, none proximally. A vastus lateralis muscle biopsy showed severe type II fiber predominance with scattered angular esterase positive fibers, also labeled intensely with NADH-TR.

## Genetic study

After obtaining negative results for Spinal Muscular Atrophy, CMT panel (PMP 22 duplication/deletion/sequencing, Cx32, MPZ, EGR2,

PRX, DGAP1, LITAF, MFN2), SOD1, SCA panel (SCA1, 2, 3, 6, 7, 8, 10, 17, DRPLA, FRDA1), and HSP panel (SPG3, SPG4, NIPA1) genetic tests, whole-exome sequencing was performed on the affected patient at the Medical Genetics Laboratories, Baylor College of Medicine as a clinical diagnostic service. Sequencing and data analyses were conducted as previously described (Yang *et al*, 2013, 2014). The exome capture reagent, VCRome 2.0, targeted approximately 20,000 genes, including the coding and UTR regions. Sequencing was performed on Illumina HiSeq 2000 platform (Illumina) with > 95% of the target exome regions covered at > 20× (average coverage > 100×). Next-generation sequencing (NGS) data were processed through the in-house-developed Mercury pipeline (Yang *et al*, 2013, 2014). Variants were classified into exonic, intronic, or intragenic, as well as their potential functional effects, and their frequencies in different populations and databases by the mentioned annotation pipelines. Variants were filtered by their observed frequencies in databases such as dbSNP, the 1000 Genomes Project, and the NHLBI Exome Sequencing Project Exome Variant Server (EVS) to filter out common polymorphisms of high frequency in healthy control populations that are likely to be benign variants. After identification of a homozygous mutation in the *SIGMAR1* gene, DNA from both of biological parents was analyzed by Sanger sequencing of the *SIGMAR1* gene, and a genomewide single nucleotide polymorphism (SNP) genotyping for homozygosity mapping was performed on the DNA sample from the father and the patient using Illumina cSNP array (Illumina, San Diego, CA).

## Antibodies

Antibodies against Sig1R(L20) (1:250, Cat# sc-16203), CYPOR(H-300) (1:250, Cat# sc-13984) were obtained from Santa Cruz Biotechnology (Santa Cruz, CA). Anti-IP$_3$R3 (1:1,000, Cat# 610313) antibody was obtained from BD biosciences (San Jose, CA). Anti-human SOD1 antibody was raised and purified in our laboratory (Bruijn *et al*, 1997). We also purchased the following commercially available antibodies: anti-IP$_3$R1 (L24/18) (1:500, Cat# 817701) (BioLegend, San Diego, CA), anti-FLAG M2 (1:1,000, Cat# F3165) (Sigma-Aldrich, St. Louis, MO), anti-c-myc (1:400, Cat# 11 667 149 001), anti-influenza hemagglutinin (HA) (1:1,000, Cat# 11 867 423 001) (both from Roche, Basel, Switzerland), anti-histone H3 (1:1,000, Cat# 9715), anti-calreticulin (1:500, Cat# 2891), anti-PDI (1:1,000, Cat# 2446), anti-VDAC (1:1,000, Cat# 4661) (Cell Signaling Technology, Danvers, MA), anti-βIII-tubulin (1:5,000, Cat# PRB-435P) (Covance, Princeton, NJ), and anti-spectrin α II (1:500, Cat# MB1622) (EMD Millipore, Billerica, MA). Alexa-conjugated secondary antibodies (1:1,000) were purchased from Life Technologies (Grand Island, NY), and horseradish peroxidase (HRP)-conjugated secondary antibodies (1:5,000) were from GE Healthcare (Waukesha, WI).

## Animals

Transgenic mice expressing wild type (B6.Cg-Tg(SOD1) 76Dpr) or mutant SOD1 (B6.Cg-Tg(SOD1-G37R) 1Dwc/J), (B6.Cg-Tg(SOD1-G85R) 148Dwc/J), (B6.Cg-Tg(SOD1-G93A) 1Gur/J) were obtained from the Jackson Laboratory (Bar Harbor, ME) or were gifts from Dr. Don Cleveland (University of California, San Diego), and

Sig1R$^{-/-}$ mice (B6.129S5-Sigmar1Gt(OST422756)Lex/Mmucd) were obtained from the Mutant Mouse Regional Resource Center (MMRRC, University of California, Davis). The Sig1R$^{-/-}$ mice were genotyped using polymerase chain reactions (PCR) with the following sense and antisense primers: 5′-ACCATGAGCTTGCCTTCTC TCG-3′, 5′-AGACCTCACTTTTCGTGGTGCC-3′, and 5′-CAGATCAAG GTCAGGAACAGATGG-3′. Genotyping for SOD1 transgenic mice was performed as previously described (Watanabe *et al*, 2014a). The mice were housed in the specific pathogen free (SPF) environment (12 h light-dark cycle; 23 ± 1°C; 50 ± 5% humidity) and treated in compliance with the requirements of the Animal Care and Use Committee of Nagoya University. The experiments using genetically modified animals and organisms were approved by the Animal Care and Use committee and the recombinant DNA experiment committee of Nagoya University (approval numbers #16013 and #143-9, respectively).

## Rotarod test, and analysis for the onset and progression of the disease

The onset of disease and end stage were determined as previously described (Watanabe *et al*, 2014a). For the rotarod test, mice were placed on the rotating rods, which accelerated from 0 to 30 or 45 rpm for 5 min (Muromachi Kikai, Tokyo, Japan). The tests were discontinued at 5 min, and the best values out of three trials were scored. An interval of 15 min was taken between each trial. For training, the mice were subjected to the same test on the day before the first rotarod test. No randomization or blinding was used in this study.

## Isolation of MAM from cultured cells and mouse tissues

MAM was isolated from cultured cells or mouse tissues as previously reported (Wieckowski *et al*, 2009; Fujimoto & Hayashi, 2011). The protein concentration in each fraction was determined by Bio-Rad Bradford protein assay kit (Bio-Rad, Richmond, CA). For total cell or tissue lysates, cells or tissues were lysed in ice-cold 50 mM Tris–HCl (pH 7.4), 150 mM NaCl, 1 mM ethylenediaminetetraacetic acid, and 1% (v/v) Triton X-100 supplemented with protease inhibitor cocktail and PhosSTOP (both from Roche). An equal amount of total protein (15 or 30 μg/lane) was separated by sodium dodecyl sulfate (SDS)–polyacrylamide gel electrophoresis and transferred on immobilon-P membrane (EMD Millipore) at 30 V, 4°C overnight without methanol as previously reported (Hayashi & Su, 2003).

## Immunofluorescence staining

Immunofluorescence staining was performed as previously described (Watanabe *et al*, 2014b) with antigen activation for Sig1R (Hayashi *et al*, 2011). In brief, the sections were activated in 10 mM Tris–HCl (pH 9.5) and 6 M urea at 85°C for 10 min. After blocking, the sections were incubated with primary antibodies overnight at 4°C. Bound antibodies were detected with Alexa-conjugated secondary antibodies. Images were obtained by confocal laser scanning microscopy (LSM-700; Carl Zeiss AG, Oberkochen, Germany) and the equipped software (Zen; Carl Zeiss AG).

## Electron microscopy

The mice were performed by Karnovsky fixative solution containing 3% glutaraldehyde and 4% paraformaldehyde in 0.1 M phosphate buffer. The lumber spinal cords were trimmed and placed in the same fixative for 3–4 h at 4°C. These specimens were post-fixed in 1% osmium tetroxide in the same buffer for 2 h at 4°C, rinsed with distilled water, block-stained overnight in a saturated solution of uranyl acetate, dehydrated in ethyl alcohol series, and embedded in epoxy resin. Following ultrathin sections cut using an ultramicrotome (UC7k; Leica Microsystems), the sections were double-stained with uranyl acetate and lead citrate, and processed for observation with a transmission electron microscope (JEM-1400Plus; JEOL, Tokyo, Japan). The circumference of each mitochondrion closely associated with ER (< 30 nm) was counted using ImageJ software as previously reported (Stoica et al, 2014).

## Cell culture

Mouse neuroblastoma Neuro2a (N2a) cells were maintained and differentiated as described previously (Watanabe et al, 2014b). Transfection was performed with Lipofectamine 2000 (Life Technologies), and plasmids encoding SOD1 and its mutants were prepared as described elsewhere (Watanabe et al, 2013, 2014b). N2a cells stably expressing human IP$_3$R3 (N2a-IP$_3$R3) were established as follows: human IP$_3$R3 cDNA, a kind gift from Dr. Katsuhiko Mikoshiba (RIKEN Brain Science Institute, Japan), was cloned with C-terminal HA tag into pcDNA3.1(+) (Life Technologies) and transfected. The cells were selected with 500 μg/ml G418 (Roche) for 2 weeks. Expression of human IP$_3$R3 in the selected clone was confirmed by immunoblotting analysis using an anti-IP$_3$R3 antibody.

## Cycloheximide chase assay, co-immunoprecipitation, and neurotoxicity assay

Cycloheximide pulse chase assay (Niwa et al, 2007) and neurotoxicity assay (Watanabe et al, 2014b) were performed as previously reported. For co-immunoprecipitation, N2a cells were seeded at $4.0 \times 10^5$ cells/well on 6-well plates. After transfection of IP$_3$R3 and FLAG tagged Sig1R, the media were replaced with the differentiation media, and the cells were incubated for 24 h. The cells were lysed in 0.5 ml of lysis buffer (50 mM Tris–HCl (pH 8.0), 150 mM NaCl, and 1% (v/v) CHAPS supplemented with protease inhibitor and PhosSTOP) and incubated with anti-FLAG M2 antibodies over-night at 4°C with gentle agitation, followed by incubation with protein G Sepharose (GE Healthcare, 15 μl) for further 1.5 h. The beads were washed four times with lysis buffer. The proteins were eluted by incubation with FLAG peptide (Sigma-Aldrich, 5 μg/ml) for 1 h at 4°C with gentle agitation.

## Ca$^{2+}$ imaging with fluo-4 or Case12-mito

siRNA against Sig1R was obtained from Life Technologies. A mitochondrial Case12-expressing plasmid (pCase12-mito) was purchased from Evrogen (Moscow, Russia). N2a-IP$_3$R3 cells were seeded at $2.0 \times 10^5$ cells/dish on poly-D-lysine-coated glass-bottom dishes (MatTek, Ashford, MA). After transfection with Lipofectamine 2000

and/or Lipofectamine RNAi MAX (Life Technologies), the cells were incubated for 24 h in the differentiation media. Fluo-4 AM (Life Technologies), PRE-084, or NE-100 (both from Sigma-Aldrich, 5 μM each) was added 1 h prior to observation. The cells were washed twice with warmed Hank's balanced saline (Life Technologies). Fluorescence intensity was measured every 3 s until 120 s, and intracellular Ca$^{2+}$ release was stimulated by adding adenosine triphosphate (ATP; 10 μM) to the media at 30 s.

## In vitro ER and mitochondria contacting assay

The assay was performed as previously reported (Sugiura et al, 2013). Briefly, pure isolated mitochondrial and microsomal fractions from N2a cells transfected with wild-type or mutant SOD1 were incubated for 15 min at 37°C in the reaction buffer (10 mM Tris–HCl (pH 7.4), 150 mM KCl, 1 mM KH$_2$PO$_4$, 5 mM MgCl$_2$, 10 mM sodium succinate). Mitochondria-associated microsomes were collected by centrifugation for 5 min at $5,000 \times g$, 4°C.

## Calpain activity assay and intracellular ATP measurement

N2a-IP$_3$R3 cells were seeded on 6-well plates at $4.0 \times 10^5$ cells/well on the day before transfection. After 6 h of transfection, the media were replaced with the differentiation media and incubated for another 24 h. An aliquot of the cell lysate (100 μg) was analyzed by calpain activity assay kit (BioVision, Mountain View, CA). To measure the level of intracellular ATP, N2a-IP$_3$R3 cells were seeded on 24-well plates at $1.0 \times 10^5$ cells/well on the day before transfection. After 6 h of transfection, the media were replaced with the differentiation media containing 1.0 g/l glucose and incubated for another 24 h. The cells ($10^4$ cells per well) were analyzed by an ATP assay kit for cells (TOYO B-net, Tokyo, Japan). The calibration was performed in $1–10^3$ fmol/well.

## Administration of PRE-084 to SOD1$^{G93A}$ mice

Mice were intraperitoneally administered 0.25 mg/kg of PRE-084 solution diluted at 16.6 μg/ml with saline three times per week (Peviani et al, 2014). As a control, the same volume of saline was administered. The administration started when the mice were 1 month old and continued until they were 3 months old.

## Statistical analysis

Statistical analysis of survival time was performed with a log-rank test. For analyses of intracellular Ca$^{2+}$ flux, fluorescence intensity was tested by two-way ANOVA with subsequent post hoc Tukey's test. The scores of rotarod test were also tested by two-way ANOVA or unpaired t-test. Suppression of Sig1R expression by siRNA was confirmed by paired t-test. All other quantified experiments were analyzed by a one-way ANOVA with subsequent post hoc Tukey's tests. The list of P-values was provided in Table EV1.

**Expanded View** for this article is available online.

## Acknowledgements

We thank the Support Unit for Animal Resource Development and Biomaterial Analysis in RIKEN BSI Research Resource Center, Center for Gene

## The paper explained

### Problem

A homozygous mutation in the gene for sigma 1 receptor (Sig1R) is a cause of inherited juvenile amyotrophic lateral sclerosis (ALS16). Sig1R localizes to the mitochondria-associated membrane (MAM), which is an interface of mitochondria and endoplasmic reticulum; however, the role of the MAM in ALS is not fully elucidated. A key unsolved question is whether disruption of the MAM is a pathomechanism shared by the other forms of ALS.

### Results

We report that ALS16-linked Sig1R variants, including a novel p.L95fs mutation, are uniformly unstable and non-functional, indicating that a loss of function of Sig1R is causative for ALS16. The onset of mutant Cu/Zn superoxide dismutase (SOD1)-mediated ALS disease in mice was accelerated when Sig1R was deficient. Moreover, Sig1R deficiency induced dissociation of the MAM components and deregulation of $Ca^{2+}$ homeostasis at the MAM through mislocalization of $IP_3R$ type 3 ($IP_3R3$), resulting in calpain activation, mitochondrial dysfunction, and neurodegeneration.

### Impact

Our study reveals that ALS16 is caused through a loss of MAM integrity induced by Sig1R deficiency, and the collapse of the MAM is a common pathomechanism in Sig1R- and SOD1-linked ALS. Moreover, our discovery of the selective enrichment of $IP_3R3$ in motor neurons suggests that the interaction of Sig1R–$IP_3R3$ at the MAM may be responsible for the selective vulnerability in ALS. Although further study is still required, our work substantially contributes to understanding the general molecular mechanism of ALS by providing evidence linking for the first time SOD1- and Sig1R-linked ALS through collapse of the MAM in motor neurons.

Research, and Center for Animal Research and Education, Nagoya University for recovery of Sig1R$^{-/-}$ mice from frozen sperm, the DNA analyses, and the animal maintenance and experiments, respectively. We also thank the patient and the family for their contribution. We also acknowledge the Division for Medical Research Engineering, Nagoya University Graduate School of Medicine, for the use of facilities including the ultramicrotome (UC7k) and electron microscope (JEM-1400Plus). This work was funded by Grants-in-aid for Scientific Research 23111006, 26293208, 16H01336 (to KY), 25860252 (to SW) from the Ministry for Education, Culture and Sports, Science and Technology, Japan, Grant-in-Aid for Research on Rare and Intractable Diseases, the Research Committee on Establishment of Novel Treatments for Amyotrophic Lateral Sclerosis, from Japan Agency for Medical Research and Development (AMED), Naito Foundation, Uehara Memorial Foundation, Japan ALS Association (JALSA) (KY), and Hori Science and Arts Foundation (SW). HT was supported by the Japan Society for the Promotion of Science (JSPS) Research Fellowships for Young Scientists. HI was supported by American Academy of Neurology Clinical Research Training Fellowship in ALS.

## Author contributions

SW and KY designed the study. SW and HN conducted the biochemical, histological, behavioral analyses with supports from OK, FE, and SJ under the supervision of KY. HT and HK performed electron microscopy. HI and PM conducted genetic analyses of the ALS patient and her family. SW and KY interpreted the data and wrote the manuscript. All authors read and approved the manuscript.

## Conflict of interest

The authors declare that they have no conflict of interest.

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
