## [Review Process File · EMBO Molecular Medicine]

Mitochondria-associated membrane collapse is a common pathomechanism in SIGMAR1- and SOD1-linked ALS

Seiji Watanabe, Hristelina Ilieva, Hiromi Tamada, Hanae Nomura, Okiru Komine, Fumito Endo, Shijie Jin, Pedro Mancias, Hiroshi Kiyama, Koji Yamanaka

Corresponding author: Koji Yamanaka, Nagoya University

Review timeline:

Submission date:	14 March 2016
Editorial Decision:	19 April 2016
Revision received:	03 August 2016
Editorial Decision:	01 September 2016
Revision received:	21 September 2016
Accepted:	27 September 2016

Transaction Report:

Editor: Roberto Buccione

1st Editorial Decision

19 April 2016

Thank you for the submission of your manuscript to EMBO Molecular Medicine.

In this case we also experienced unusual difficulties in securing three willing and appropriate reviewers. As a further delay cannot be justified I have decided to proceed based on the two available consistent evaluations.

Both Reviewers are generally positive on the overall relevance and importance of the message conveyed by your manuscript although they express fundamental concerns that require significant action. I will not dwell into much detail as their comments are clear. I would like, however, to highlight a few general points.

The main concerns, which are for the most overlapping, centre upon the lack of strong evidence for a number of points including actual localisation of IP3R3 and SOD1 at MAMs and its disease relevance and the role of the Sig1R-IP3R3 interaction. Numerous experimental improvements are required, including better and more appropriate controls.

I should also mention that Reviewer 2 notes the perfunctory acknowledgement of the previous HMG study on Sig1R KO mice; I had noticed this too and agree. We are especially concerned that prior work is properly acknowledged and referenced.

In conclusion, while publication of the paper cannot be considered at this stage, we would be willing to consider a substantially revised submission, with the understanding that the Reviewers' concerns must be addressed with additional experimental data where appropriate.

Please note that it is EMBO Molecular Medicine policy to allow a single round of revision only and that, therefore, acceptance or rejection of the manuscript will depend on the completeness of your responses and on the outcome of the required experimentation included in the next, final version of the manuscript.

As you might know, EMBO Molecular Medicine has a "scooping protection" policy, whereby similar findings that are published by others during review or revision are not a criterion for rejection. However, I do ask you to get in touch with us after three months if you have not completed your revision, to update us on the status. Please also contact us as soon as possible if similar work is published elsewhere.

Please note that EMBO Molecular Medicine now requires a complete author checklist (<http://embomolmed.embopress.org/authorguide#editorial3>) to be submitted with all revised manuscripts. Provision of the author checklist is mandatory at revision stage; The checklist is designed to enhance and standardize reporting of key information in research papers and to support reanalysis and repetition of experiments by the community. The list covers key information for figure panels and captions and focuses on statistics, the reporting of reagents, animal models and human subject-derived data, as well as guidance to optimise data accessibility.

I also suggest that you carefully adhere to our guidelines for publication in your next version, including presentation of statistical analyses and our new requirements for supplemental data (see also below) to speed up the pre-acceptance process in case of a favourable outcome.

Please note that we now mandate that all corresponding authors list an ORCID digital identifier. You may do so through our web platform upon submission and the procedure takes <90 seconds to complete. We also encourage co-authors to supply an ORCID identifier, which will be linked to their name for unambiguous name identification.

I look forward to seeing a revised form of your manuscript as soon as possible.

***** Reviewer's comments *****

Referee #1 (Remarks):

In this study, Watanabe and colleagues investigate the pathogenesis of ALS16 (due to mutation in the Sig1R). The Sig1R has been reported to localize to MAMs, but the role of these is ALS is unclear. Here, the authors have identified a new mutation in a single ALS patient (p.L95fs, due to uniparental disomy). The authors show that this mutation and the previously reported p.E102Q variant are degraded in cells, and thus conclude the disorder stems from a loss of Sig1R. They tested Sig1R for interaction with another MAM protein, IP3R, and found that depletion of Sig1R caused an increase in Ca²⁺ flux into cytoplasm, which could be rescued by WT but not mutant Sig1R. The onset of mutant SOD1-related ALS was accelerated in a mouse model when Sig1R was deficient. The authors conclude that mislocalization of IP3R3 from the MAM causes increased Ca²⁺ flux to the cytoplasm, with calpain activation and mitochondrial dysfunction, linking these forms of ALS by MAM collapse.

ALS pathogenesis is an area of great interest and active research, and there are some important points made here, but I have a few issues to be addressed:

1. More details should be provided on the whole exome analysis that identified the SIGMAR1 gene mutation (i.e., coverage, platform, filters used for analysis, etc.).
2. Re: the p.L95fs patient. Is Sig1R expressed in skin fibroblasts? If so, it would be important to perform RT-PCR to see if the mRNA is degraded and/or immunoblot for endogenous Sig1R to support further the loss-of-function mechanism proposed.
3. Regarding Fig. 2D. Is there a cell type where the authors can show that endogenous IP3R3 is so highly localized to the MAM fraction, since this is a key point of the paper. As it stands, the finding in Fig. 2D is robust, but also represents localization of an overexpressed protein in a cell type that does not normally express it.

4. Regarding Fig. 4B. I was surprised to see calreticulin so specifically localized to MAMs, since numerous localization studies show this to be broadly expressed in ER. Perhaps one or two other ER proteins could strengthen the validity of the fractionation (e.g., an ER-shaping reticulon or REEP protein).

5. Regarding Fig. 6. Does the mislocalization of IP3R3 in the setting of mutant SOD1 or Sig1R loss merely reflect an alteration in general ER morphology? This could be assessed using other ER markers.

Minor points:

1. The Bernard-Marissal et al. reference on page 28 is incomplete.
2. The Hayashi et al 2007a and 2007b references are duplicated on page 29, and are in fact the same publication.

Referee #2 (Remarks):

This paper looks at the mechanisms underlying ALS16 and SOD1 ALS. That Sig1R1 regulates MAM has been published (e.g. Bernard-Marissal et al., HMG 2015) but the link to SOD1 is new and intriguing.

The main concerns I have with the manuscript are:

1: There is no direct evidence that MAM is actually disrupted in Sig1R or SOD1 cells/mice. Without that direct proof e.g. by EM. Hence I don't think the authors can conclude there is a "collapse of MAM" or that "Sig1R is crucial for integrity of the MAM".

2: There is no direct evidence that IP3R3 is crucial for neurodegeneration - only a correlation. Hence the conclusion that "integrity of the MAM is crucial for the selective vulnerability in ALS" is too strong.

Overall comments:

All blots need molecular weight markers
The manuscript would benefit of careful proofreading.

Specific comments:

Introduction - the authors gloss over the study showing SIG1R KO mice have reduced ER-mitochondria interaction etc. published in HMG last year. They should also include the published work showing reduced MAM in TDP-43 associated ALS.

Figure 2:

Panel D: Markers showing the purity of the fractions need to be included; why is there a significant amount of IP3R in the nuclear fraction?

Panel E: IP3R3 is coming down with anti-Flag Ab. Hence no conclusions can be made. That aside the conclusion of the authors is incorrect - IP3R3 is found in all fractions. This experiment needs to be removed and replaced.

Figure 3:

This data would be much stronger if wtSOD1/Sig1R were to be included. At least the authors should provide more information on their Sig1R mice. How do they know that SOD1G85R is not exacerbating the Sig1R KO phenotype? or that the effects are merely additive?

Figure 4:

I don't believe the authors can conclude that SOD1 specifically accumulates in MAM, there is just as much accumulation in the Nuclear and Mitochondrial fractions. In B, most of the G85R is actually in the nuclear fraction. From which samples do the fraction controls derive? there should be controls for each fractionation.

The top panel in C is the same panel as in B!

Panel D: How is it possible to have PDI in samples where there are no mitochondria - does this

show the PDI signal spins down independently of mitochondria? In that case there is no specific association of ER with mitochondria. These blot do not support the conclusions.

Figure 5:

Panel A, B C, D:

Clarification is needed on how these blots were obtained - are they from one mouse or several?

A loading control is needed for each fraction, and a control for the purity of the fractions.

What happens to wtSOD1 in the same assays or mouse SOD1?

Panel C, D: The results in G37R are the opposite of G93A or G85R (Increase v decrease) but the text says they are the same?

Panel E and F:

Are these blots from the same mouse? A loading control is needed for each fraction, and a control for the purity of the fractions.

Figure 6:

Antibody controls are needed to show there is no crosstalk.

The statement "there is complete colocalization" is not supported by this data without any quantification.

The conclusion that "disruption of Sig1R-IP3R interaction is involved in degeneration" is too strong - there is a correlation, but no direct proof.

The conclusion that "Sig1R is crucial for integrity of the MAM" is too strong - direct proof by e.g. EM is needed.

Figure 7:

Panel C and F are lacking the non-G85R transfected control to which the other conditions should be normalized.

Discussion:

What do the authors mean with "These studies imply that mutant SOD1 actively accumulates at the MAM for its degradation. Increasing amounts of mutant SOD1 in a pre-symptomatic stage may indicate that the intracellular degradation systems at the MAM are overwhelmed." What is the evidence that proteins are degraded at MAM?

The authors should compare their data to Bernard-Marissal et al. (HMG 2015) in more detail. Some discussion of MAM in TDP43 ALS, AD and PD could help placing this study in a wider context.

1st Revision - authors' response

03 August 2016

Response to the referees: Watanabe et al.

Referee #1 (Remarks):

In this study, Watanabe and colleagues investigate the pathogenesis of ALS16 (due to mutation in the Sig1R). The Sig1R has been reported to localize to MAMs, but the role of these in ALS is unclear. Here, the authors have identified a new mutation in a single ALS patient (p.L95fs, due to uniparental disomy). The authors show that this mutation and the previously reported p.E102Q variant are degraded in cells, and thus conclude the disorder stems from a loss of Sig1R. They tested Sig1R for interaction with another MAM protein, IP3R, and found that depletion of Sig1R caused an increase in Ca²⁺ flux into cytoplasm, which could be rescued by WT but not mutant Sig1R. The onset of mutant SOD1-related ALS was accelerated in a mouse model when Sig1R was deficient. The authors conclude that mislocalization of IP3R from the MAM causes increased Ca²⁺ flux to the cytoplasm, with calpain activation and mitochondrial dysfunction, linking these forms of ALS by

MAM collapse. ALS pathogenesis is an area of great interest and active research, and there are some important points made here, but I have a few issues to be addressed:

Response: We are so grateful for the positive evaluation of this reviewer on our work.

1. *More details should be provided on the whole exome analysis that identified the SIGMAR1 gene mutation (i.e., coverage, platform, filters used for analysis, etc.).*

Response: According to the reviewer's suggestion, we added more details on the whole exome analysis in the *Materials and Methods* section.

2. *Re: the p.L95fs patient. Is Sig1R expressed in skin fibroblasts? If so, it would be important to perform RT-PCR to see if the mRNA is degraded and/or immunoblot for endogenous Sig1R to support further the loss-of-function mechanism proposed.*

Response: Unfortunately, the cells from the patient are not readily accessible. L95fs mutant protein lacks most of the cytoplasmic portion including ligand-binding motifs (Fig 1B), is very unstable (Fig 2B), and is unable to control Ca²⁺ flux (Fig 2I-L). Therefore, together with evidence for the analysis of E102Q mutant in our manuscript, we propose that loss-of-function mechanism is responsible for SigR1-linked ALS. We hope that the reviewer understands our situation.

3. *Regarding Fig. 2D. Is there a cell type where the authors can show that endogenous IP₃R3 is so highly localized to the MAM fraction, since this is a key point of the paper. As it stands, the finding in Fig. 2D is robust, but also represents localization of an overexpressed protein in a cell type that does not normally express it.*

Response: We agree that this is a very important point. To confirm our initial finding, we performed the new experiments by using two cell lines; HeLa cells, which endogenously express IP₃R3, and Neuro2a (N2a) cells stably expressing human IP₃R3 at a low level. The results are shown in revised Fig 2E and F. Compared with IP₃R1, IP₃R3 is highly enriched at the MAM in both the cells, indicating that IP₃R3 is localized at the MAM in a normal physiological condition. Our result is consistent to the one shown by Hayashi and Su (Cell, 2007). We interpreted some contamination in the other fraction of IP₃R3 in HeLa cells as a technical limitation of fractionation methods.

4. *Regarding Fig. 4B. I was surprised to see calreticulin so specifically localized to MAMs, since numerous localization studies show this to be broadly expressed in ER. Perhaps one or two other ER proteins could strengthen the validity of the fractionation (e.g., an ER-shaping reticulon or REEP protein).*

Response: According to the reviewer's suggestion, we re-performed the MAM isolation in new Fig 2E and F, and found that calreticulin was also localized in P3 fractions in the most cases. Our previous results may be due to a poor sensitivity of the immunoblotting. Therefore, we replaced the immunoblotting images of calreticulin, and added the immunoblots of protein disulfide isomerase (PDI) to strengthen the validity of the fractionation in new Fig. 2E, 2F, 4A and 4B.

5. *Regarding Fig. 6. Does the mislocalization of IP₃R3 in the setting of mutant SOD1 or Sig1R loss merely reflect an alteration in general ER morphology? This could be assessed using other ER markers.*

Response: To address the reviewer's point, we performed immunostaining with anti-PDI antibody, a MAM-enriched ER marker in the spinal cord section of Sig1R^{-/-} mice (Fig EV3A). The general morphology of ER was not affected by Sig1R deficiency. Therefore, we interpreted that MAM disruption is probably more responsible for the mislocalization of IP₃R3 than an alteration in general ER morphology in the revised article (Page 11).

Minor points:

1. *The Bernard-Marissal et al. reference on page 28 is incomplete.*

2. *The Hayashi et al 2007a and 2007b references are duplicated on page 29, and are in fact the same publication.*

Response: We corrected the errors in references. We thank to the reviewer.

Referee #2 (Remarks):

This paper looks at the mechanisms underlying ALS16 and SOD1 ALS. That Sig1R1 is regulates MAM has been published (e.g. Bernard-Marissal et al., HMG 2015) but the link to SOD1 is new and intriguing.

Response: We are so grateful for the positive evaluation on the link between SOD1- and Sig1R-ALS.

Note: We found that "Bernard-Marissal et al. HMG 2015", which the reviewer indicated, was, in fact, "Bernard-Marissal et al. Brain 2015". Therefore, we regarded the study as latter where it was indicated in our response or in the revised manuscript.

The main concerns I have with the manuscript are:

1: There is no direct evidence that MAM is actually disrupted in Sig1R or SOD1 cells/mice. Without that direct proof e.g. by EM. Hence I don't think the authors can conclude there is a "collapse of MAM" or that "Sig1R is crucial for integrity of the MAM".

Response: We agree with the reviewer. In this revised article, we performed electron microscopy to provide the direct evidence of the MAM disruption. As shown in Figs 6J, 6K and EV3B, the ER-mitochondria association was apparently reduced in both SOD1^{G85R} and Sig1R^{-/-} mouse motor neurons. These results are very similar to that of the previous study demonstrating MAM disruption in a VAPB-linked ALS cell model (Stoica et al, Nature Communications, 2014). We interpreted that this is the first direct evidence for the MAM disruption in SOD1-linked ALS. The discussion of this point has also been revised (Page 14).

2: There is no direct evidence that IP₃R3 is crucial for neurodegeneration - only a correlation. Hence the conclusion that "integrity of the MAM is crucial for the selective vulnerability in ALS" is too strong.

Response: We agree that there is not a direct evidence of IP₃R3 involvement in motor neuron degeneration in our current study. However, we performed *in vitro* experiment indicating that IP₃R3 dysregulation is involved in the cell death mediated by mutant SOD1. Especially, IP₃R3 expression in N2a cells exacerbated the cytotoxicity of mutant SOD1 (revised Fig 7A). In addition to this, Sig1R modulated IP₃R3-mediated Ca²⁺ regulation (Fig 2I and J), and affects calpain activation and ATP synthesis (Figs 7 and 8). We consider that these data are, even though they are indirect evidence, sufficient to claim IP₃R3 is "involved" in the neurodegeneration, but as the reviewer said, it seems to be too strong to claim IP₃R3 is "crucial". Therefore, the text regarding this point has been revised (page 11, page12, a legend for figure 7).

Overall comments:

All blots need molecular weight markers

The manuscript would benefit of careful proofreading.

Response: We added molecular weight markers to each blot, and the manuscript was revised. We apologize for our grammatical errors in the initial version.

Specific comments:

Introduction - the authors gloss over the study showing SIG1R KO mice have reduced ER-mitochondria interaction etc. published in HMG last year. They should also include the published work showing reduced MAM in TDP-43 associated ALS.

Response: According to the reviewer's suggestion, we revised the manuscript to cite and summarize the indicated recent works (Bernard-Marissal et al, Brain, 2015, Stoica et al, Nat Commune, 2014, Prause et al, Hum Mol Genet, 2013) in the introduction section (page 4). During the revision process, novel recessive mutations in *SIGMAR1* gene have been reported as a cause of distal hereditary motor neuropathy (dHMN) (Gregianin et al. Hum Mol Genet, 2016). Their work was also cited in the introduction, and these mutations were also included in new Figure 1B.

Figure 2:

Panel D: Markers showing the purity of the fractions need to be included; why is there a significant amount of IP₃R in the nuclear fraction?

Response: We replaced old Fig 2D with Fig 2E and F to include the immunoblots to warrant the purity of the fractions, according to the comments from reviewers #1 and #2. In revised Fig 2D, we added the schematic outline of the fractionation. P1 (nuclei and debris; indicated as a nuclear fraction in the previous version) indicates a resulting pellet of centrifugation at 600 x g. Since IP₃Rs were membrane proteins and not highly soluble partly due to their large size (over 300 kDa),

incompletely solubilized IP₃Rs were detected as debris in P1 fractions. To clarify this point, we renamed the fractions as P1 (nuclei and debris) instead of nuclear fractions, and replaced old Fig 2D.

Panel E: IP₃R3 is coming down with anti-Flag Ab. Hence no conclusions can be made. That aside the conclusion of the authors is incorrect – IP₃R is found in all fractions. This experiment needs to be removed and replaced.

Response: We agree with the reviewer's point, and replace the old Fig 2E with the revised Fig 2G. Since L95fs Sig1R variant was very unstable, we eliminated the L95fs variants from this experiment and performed immunoprecipitation of FLAG-tagged wild-type or E102Q Sig1R with an anti-FLAG antibody. In this experiment, we changed the detergents and found that using CHAPS instead of NP-40 greatly reduced the non-specific binding of IP₃R3. In our revised experiment, we concluded that E102Q Sig1R variant was non-functional, since this mutant protein was unable to interact with IP₃R3 (page 7).

Figure 3:

This data would be much stronger if wtSOD1/Sig1R were to be included. At least the authors should provide more information on their Sig1R mice. How do they know that SOD1G85R is not exacerbating the Sig1R KO phenotype? or that the effects are merely additive?

Response: According to the reviewer's suggestion, we added more information of Sig1R^{-/-} mice. As shown in the revised Fig 3B and C, Sig1R deficiency itself did not show any severe neurological phenotypes in mice, i.e. it did not lose the body weight or affect the lifespan approximately until 400 days, which was beyond the mean survival time of SOD1^{G85R} mice. Sig1R^{-/-} mice did not show paralysis like SOD1^{G85R}, however, showed age-dependent moderate decline of motor performance measured by a rotarod test (Fig 3G and H).

Figure 4:

I don't believe the authors can conclude that SOD1 specifically accumulates in MAM, there is just as much accumulation in the Nuclear and Mitochondrial fractions. In B, most of the G85R is actually in the nuclear fraction.

Response: Mutant SOD1 proteins, especially SOD1^{G85R}, forms insoluble aggregates as previously reported. Since the nuclear fraction (now renamed as P1 to avoid confusion) contains insoluble debris, insoluble SOD1^{G85R} aggregates were found in this fraction. In addition to this, it is also reported that mutant SOD1 binds to mitochondria outer membranes (Israelson et al, Neuron 2010). In this study, we revealed that SOD1 was accumulated not only in the indicated fractions (P1 and mitochondria) but also in the MAM fraction for the first time.

The corresponding text was also appropriately revised as follows.

“Mutant SOD1 proteins are accumulated at the MAM and mitochondria in the cultured neurons and affected tissues

To investigate the mechanisms through which Sig1R deficiency exacerbated the disease of mutant SOD1 mice, we performed the subcellular fractionation from mutant SOD1-expressing N2a cells and SOD1 transgenic mouse tissues as shown in Fig 2D. In contrast to the cytosolic localization of SOD1^{WT} protein, we observed mutant SOD1 was partially localized in the mitochondrial fraction in N2a cells and the spinal cords of end-stage mutant SOD1 mice as previously reported (Figs 4A and B) (Liu et al., 2004). In addition, we found that mutant SOD1 was accumulated also at the MAM both in N2a cells and in the spinal cord of end-stage mutant SOD1 mice (Figs 4A and B).” (page 8-9)

From which samples do the fraction controls derive? there should be controls for each fractionation.

Response: According to the reviewer's comment, fraction control data for each mutant was presented in Fig EV1, and we indicated that fraction control data in Fig 4 is derived from N2a cells transfected with SOD1^{WT} (Fig 4A) or SOD1^{WT} mouse spinal cords (Fig 4B).

The top panel in C is the same panel as in B!

Response: We deeply apologize our mistake. We replaced the figure panel with a correct blot (revised Fig 4C, top panel).

Panel D: How it is possible to have PDI in samples where there are no mitochondria - does this show the PDI signal spins down independently of mitochondria? In that case there is no specific association of ER with mitochondria. These blot do not support the conclusions.

Response: In Fig 5D, the four lanes from the left were the purified fractions which were used in the pull-down experiments, indicating pure mitochondria and ER (now labeled as input). Then, the pellets from the pull-down experiments were resolved and loaded on the last four lanes. The data from the last four lanes indicate that ER-mitochondria association *in vitro* was predominantly seen in the presence of wild-type mitochondria (Fig 4D, lane 5 and 6). We revised labeling of the figure to make it clear which samples are loaded on each lane. We are sorry for the insufficient information in the initial version of the figure.

Figure 5:

Panel A, B C, D:

Clarification is needed on how these blots were obtained - are they from one mouse or several?

Response: In Fig 5, we loaded fractions from a single mouse tissue on each lane, and the blotting images were the representative ones in at least three independent experiments. Therefore, according to the reviewer's comment, we added the description to the Fig 5 legend as following:

"Fractions from a single mouse tissue at the indicated ages were loaded on each lane. Representative blots from at least three independent experiments were shown."

A loading control is needed for each fraction, and a control for the purity of the fractions.

What happens to wtSOD1 in the same assays or mouse SOD1?

Response: We added confirmation data of the fractionation in Fig EV1. For SOD1^{WT} protein, we showed that SOD1^{WT} is localized specifically in the cytosolic fraction both in transfected N2a cells (Fig 4A, top) and the spinal cords of SOD1^{WT} transgenic mice (Fig 4B, top).

Panel C, D: The results in G37R are the opposite of G93A or G85R (Increase v decrease) but the text says they are the same?

Response: We agree the points, and revised the texts as follows.

"Accumulation of SOD1^{G93A} at the MAM was observed at a pre-symptomatic stage but temporarily reduced around the onset of the disease (Figs 5A and B). Similar time courses were observed in SOD1^{G85R} (Fig 5C), while reduction of SOD1^{G37R} level at the MAM was observed after the disease onset (Fig 5D)." (Page 9)

Although there is a variation of the time when reduction of mutant SOD1 at the MAM was observed, we interpret that disruption of the MAM may be involved in the onset or progression of the disease.

Panel E and F:

Are these blots from the same mouse? A loading control is needed for each fraction, and a control for the purity of the fractions.

Response: In this revised manuscript, we revised the figure legend to describe that each lane represents the result of a single mouse tissues, and that the images were representative ones from three independent experiments as described above (see the response to the reviewer's comments on Fig 5 panel A-D). Because all the MAM enriched proteins we have tested were decreased from the MAM fraction in SOD1^{G93A} spinal cords, it was difficult to select the proper loading controls. Therefore, we adjusted the amount of total proteins by Bradford assay. To verify the results of Bradford assay, we used GAPDH blots using cytoplasmic fractions as a loading control. We hope the reviewer understands our experimental condition and accepts our approach.

Figure 6:

Antibody controls are needed to show there is no crosstalk.

The statement "there is complete colocalization" is not supported by this data without any quantification.

Response: To avoid the possibility of crosstalk of two antibodies, we performed the single staining with anti-Sig1R or anti-IP₃R3 antibody, respectively, and obtained similar results to the ones with a double staining (see attached supplementary figure for the reviewer). In addition to this, the colocalization of IP₃R3 and Sig1R is also confirmed in the other cited studies (Bernard-Marissal et al, Brain, 2015; Hayashi & Su, Cell, 2007). Still, we agree that "complete colocalization" should be supported by the quantified data. Therefore, we eliminated the word "complete" from the revised manuscript.

The conclusion that "disruption of Sig1R-IP3R3 interaction is involved in degeneration" is too strong - there is a correlation, but no direct proof.

The conclusion that "Sig1R is crucial for integrity of the MAM" is too strong - direct proof by e.g. EM is needed.

Response: We have already addressed this point above in the main concern #1 by this reviewer (Fig 6J, K and Fig EV3).

Figure 7:

Panel C and F are lacking the non-G85R transfected control to which the other conditions should be normalized.

Response: We added the results of wild-type SOD1 transfected controls, and normalized other conditions by the data of SOD1^{WT} transfected control. Revised data are now included in revised Fig 7D and G.

Discussion:

What do the authors mean with "These studies imply that mutant SOD1 actively accumulates at the MAM for its degradation. Increasing amounts of mutant SOD1 in a pre-symptomatic stage may indicate that the intracellular degradation systems at the MAM are overwhelmed." What is the evidence that proteins are degraded at MAM?

Response: Recent studies, which we cited in the manuscript, demonstrated that the proteins involved in proteostatic pathways are accumulated at the MAM: MITOL is an E3 ubiquitin-ligase, PERK and BiP are the ER stress sensors, and the autophagy starts at the MAM. Therefore, we discussed the possibility that the misfolded proteins may be recognized at the MAM to trigger a response against the misfolded protein-induced stress. To make this point clear, we revised the manuscript in page 14-15.

The authors should compare their data to Bernard-Marissal et al. (HMG 2015) in more detail. Some discussion of MAM in TDP43 ALS, AD and PD could help placing this study in a wider context.

Response: We appreciate the reviewer's comment. According to the comments, we revised the discussion and added two novel paragraphs: one is discussing about the previous study by Bernard-Marissal et al. (Bernard-Marissal et al, Brain 2015) (Page 18), and the other is discussing about role of the MAM in Alzheimer's disease or Parkinson's disease (Page 18-19).

Supplementary Figure for the reviewers

5 months-old non-transgenic mouse spinal cord (ventral horn)

1st: α IP₃R3

2nd: Alex488- α mouse IgG

1st: α Sig1R

2nd: Alex488- α goat IgG

Scale bars: 50 μ m

Immunofluorescent staining was performed as described in our manuscript. To avoid the cross reactivity of the antibodies, the slide was independently stained with each indicated antibody.

References cited in the response to the reviewers

Bernard-Marissal N, Medard JJ, Azzedine H, Chrast R (2015) Dysfunction in endoplasmic reticulum-mitochondria crosstalk underlies SIGMAR1 loss of function mediated motor neuron degeneration. Brain 138: 875-890.

Hayashi T, Su TP (2007) Sigma-1 receptor chaperones at the ER-mitochondrion interface regulate Ca(2+) signaling and cell survival. *Cell* 131: 596-610

Israelson A, Arbel N, Da Cruz S, Ilieva H, Yamanaka K, Shoshan-Barmatz V, Cleveland DW (2010) Misfolded mutant SOD1 directly inhibits VDAC1 conductance in a mouse model of inherited ALS. *Neuron* 67: 575-587

Prause J, Goswami A, Katona I, Roos A, Schnizler M, Bushuven E, Dreier A, Buchkremer S, Johann S, Beyer C et al (2013) Altered localization, abnormal modification and loss of function of Sigma receptor-1 in amyotrophic lateral sclerosis. *Human molecular genetics* 22: 1581-1600

Stoica R, De Vos KJ, Paillusson S, Mueller S, Sancho RM, Lau KF, Vizcay-Barrena G, Lin WL, Xu YF, Lewis J et al (2014) ER-mitochondria associations are regulated by the VAPB-PTPIP51 interaction and are disrupted by ALS/FTD-associated TDP-43. *Nature communications* 5: 3996

2nd Editorial Decision

01 September 2016

Thank you for the submission of your revised manuscript to EMBO Molecular Medicine. We have now received the enclosed reports from the referees that were asked to re-assess it. As you will see the reviewers are now globally supportive and I am pleased to inform you that we will be able to accept your manuscript pending the following final issues and amendments:

- 1) Please incorporate the changes requested by reviewer #2
- 2) Every published paper now includes a 'Synopsis' to further enhance discoverability. Synopses are displayed on the journal webpage and are freely accessible to all readers. They include a short standfirst as well as 2-5 one sentence bullet points that summarise the paper. Please provide the synopsis including the short list of bullet points that summarise the key NEW findings. The bullet points should be designed to be complementary to the abstract - i.e. not repeat the same text. We encourage inclusion of key acronyms and quantitative information. Please use the passive voice. Please attach this information in a separate file or send them by email, we will incorporate it accordingly. You are also welcome to suggest a striking image or visual abstract to illustrate your article. If you do please provide a jpeg file 550 px-wide x 400-px high.
- 3) As per our Author Guidelines, the description of all reported data that includes statistical testing must state the name of the statistical test used to generate error bars and P values, the number (n) of independent experiments underlying each data point (not replicate measures of one sample), and the actual P value for each test (not merely 'significant' or 'P < 0.05'). You may include the P values in the figure legends.
- 4) The manuscript must include a statement in the Materials and Methods identifying the institutional and/or licensing committee approving the experiments, including any relevant details (like how many animals were used, of which gender, at what age, which strains, if genetically modified, on which background, housing details, etc). We encourage authors to follow the ARRIVE guidelines for reporting studies involving animals. Please see the EQUATOR website for details: <http://www.equator-network.org/reporting-guidelines/improving-bioscience-research-reporting-the-arrive-guidelines-for-reporting-animal-research/>. Please make sure that ALL the above details are reported.
- 5) We encourage the publication of source data, particularly for electrophoretic gels and blots, with the aim of making primary data more accessible and transparent to the reader. While this is not compulsory, in this case, given the potential image issues mentioned above, it would be advisable to provide a PDF file per figure that contains the original, uncropped and unprocessed scans of the gels used in the manuscript. The PDF files should be labeled with the appropriate figure/panel number, and should have molecular weight markers; further annotation may be useful but is not essential. The PDF files will be published online with the article as supplementary "Source Data" files. If you have any questions regarding this just contact me.

Please submit your revised manuscript within two weeks. I look forward to seeing a revised form of

your manuscript as soon as possible.

I look forward to reading a new revised version of your manuscript as soon as possible.

***** Reviewer's comments *****

Referee #1 (Comments on Novelty/Model System):

Design appears appropriate for the mechanistic studies described.

Referee #1 (Remarks):

The authors have addressed each of the issues I raised in my last review.

Referee #2 (Remarks):

The authors have added a substantial amount of novel data that answers my previous comments and strengthens the paper considerably.

Two minor points:

On page 15 the authors write "...VAPB-linked ALS cell model (Stoica et al. 2014),...". This should be TDP-43? The correct reference for VAPB is De Vos et al HMG 2012.

Stoica et al have now published reduced ER/Mitochondria contacts in FUS-related ALS (EMBO Rep). This reference could be added in the discussion?

In the revised figure 4D, there is no control ER only control to show PDI labeled membranes do not spin down at 5000g without addition of mitochondria.

2nd Revision - authors' response to editor

21 September 2016

We are pleased to hear that our manuscript is acceptable in principle pending the final issues and amendments suggested by the editor.

In this final revised version, we revised our manuscript according to the points raised by the editor as below.

1. Main text

1) The changes requested by reviewer #2 are now reflected. We revised the *introduction* and *discussion* sections to add the references mentioned by the referee: 1. De Vos et al. *Hum Mol Genet* (2012), 2. Stoica et al. *EMBO Rep* (2016).

2) Antibody dilution is now added to "Antibodies" paragraph in the *material & method* section.

3) The methods used for statistical analysis were described in each of the figure legends.

4) All of the actual *p*-value were provided in the Table EV1, since they were too many to include in the legends. In Figs 2H, 3G and H, the actual *p*-values were added in the figures.

5) The description and approval numbers for the institutional review board for medical research, the Animal Care and Use committee, and the Recombinant DNA Advisory Committee of Nagoya University were added in the *material & method* section.

2. Figures

1) A full set of source data was provided as supplementary information (source data for figures 2-7, EV1).

We often cut the immunoblotting membrane prior to incubation with primary antibodies, allowing us to examine the expressions for multiple molecules with different molecular weights in the same experimental condition. Therefore, some excised membranes were included in the source data.

2) Indicators of the "significance" (asterisks (*)) and daggers (†) were amended to represent the actual *p*-values in Fig 2B, 3F and 6K.

3) Incorrect labeling in Fig 5A ("Nuc" and "Micro") was replaced ("P1" and "P3", respectively).

4) We found the magnification was incorrect in Fig 6J (Non-Tg). Therefore, we revised the figure to adjust the magnifications.

In addition, the response to the referee #2 was attached.

We believe that the findings of this study are relevant to the scope of *EMBO Molecular Medicine*, and will be of interest to its broad readership. This manuscript has not been published elsewhere, and is not under consideration by other journals. All the authors have approved the manuscript and agreed with submission to your prestigious journal.

Thank you for your prompt consideration of our manuscript.

2nd Revision - authors' response to reviewers

21 September 2016

Response to the referees: Watanabe et al.

Referee #2 (Remarks):

The authors have added a substantial amount of novel data that answers my previous comments and strengthens the paper considerably.

Response: We are so grateful for the positive evaluation of this reviewer on our revision.

Two minor points:

On page 15 the authors write "...VAPB-linked ALS cell model (Stoica et al. 2014),...". This should be TDP-43? The correct reference for VAPB is De Vos et al HMG 2012.

Stoica et al have now published reduced ER/Mitochondria contacts in FUS-related ALS (EMBO Rep). This reference could be added in the discussion?

Response: We appreciate for the correction. According to the reviewer's comments, we added the above mentioned references in the introduction and discussion sections (pages 4, 14).

In the revised figure 4D, there is no control ER only control to show PDI labeled membranes do not spin down at 5000g without addition of mitochondria.

Response: In the experiment shown in Fig 4D, the most important finding is that the more ER membranes were pelleted with mitochondria from the SOD1^{WT} transfected cells than with mitochondria from SOD1^{G85R} transfected ones. Even when a trace amount of ER membranes was pelleted by a middle-speed centrifugation (5,000×g), this does not affect our interpretation that SOD1^{G85R} mitochondria showed less binding ability to ER compared to SOD1^{WT} mitochondria. Therefore, we chose not adding such controls. However, if the editor feels this is mandatory, we will perform it.

Corresponding Author Name: Koji Yamanaka

Journal Submitted to: EMBO molecular medicine

Manuscript Number: EMM-2016-06403